# Viral N protein hijacks deaminase-containing RNA granules to enhance SARS-CoV-2 mutagenesis

Zhean Li[1,2,11], Lingling Luo[3,4,11], Xiaohui Ju [5,11], Shisheng Huang[6,11], Liqun Lei[1], Yanying Yu[5], Jia Liu [7], Pumin Zhang[1], Tian Chi [6], Peixiang Ma [8,9], Cheng Huang [3✉], Xingxu Huang [1,6✉], Qiang Ding [5✉] & Yu Zhang [6,10✉]

## Abstract

Host cell-encoded deaminases act as antiviral restriction factors to impair viral replication and production through introducing mutations in the viral genome. We sought to understand whether deaminases are involved in SARS-CoV-2 mutation and replication, and how the viral factors interact with deaminases to trigger these processes. Here, we show that APOBEC and ADAR deaminases act as the driving forces for SARS-CoV-2 mutagenesis, thereby blocking viral infection and production. Mechanistically, SARS-CoV-2 nucleocapsid (N) protein, which is responsible for packaging viral genomic RNA, interacts with host deaminases and co-localizes with them at stress granules to facilitate viral RNA mutagenesis. N proteins from several coronaviruses interact with host deaminases at RNA granules in a manner dependent on its F17 residue, suggesting a conserved role in modulation of viral mutagenesis in other coronaviruses. Furthermore, mutant N protein bearing a F17A substitution cannot localize to deaminase-containing RNA granules and leads to reduced mutagenesis of viral RNA, providing support for its function in enhancing deaminase-dependent viral RNA editing. Our study thus provides further insight into virus-host cell interactions mediating SARS-CoV-2 evolution.

**Keywords** Innate Immunity; Deaminases; SARS-CoV-2; Deaminases; Mutagenesis
**Subject Categories** Microbiology, Virology & Host Pathogen Interaction; RNA Biology

## Introduction

SARS-CoV-2, a member of the subfamily Coronavirinae within the family Coronaviridae, which belongs to the order Nidovirales, is a positive-sense, single-strand (ss) RNA virus. The continuous emergence of SARS-CoV-2 variants of concern (VOC) with altered transmissibility, antigenicity and pathogenicity poses a challenge for vaccine development. To tackle this problem, it is necessary to elucidate the drivers of viral evolution and track the possible future evolutionary trajectories (Collie et al, 2022; Dejnirattisai et al, 2022; Schmidt et al, 2022). In general, viral RNA mutations emerge from two sources, spontaneous random replication errors made by RNA-dependent RNA polymerase (RdRP), and host-driven viral genome mutations, such as those induced by host-encoded deaminases and reactive oxygen species (ROS) (Di Giorgio et al, 2020; Wang et al, 2020a; Zhang et al, 2021). While host-driven mutations are an integral component of the innate antiviral response, capable of blocking viral replication, viruses can also exploit these mutations to evolve variants with improved adaptation and fitness (Jern et al, 2009; Monajemi et al, 2012; Sheehy et al, 2003; Turelli et al, 2004).

Human deaminases, including the adenosine deaminases acting on RNA (ADARs) and the apolipoprotein-B (*ApoB*) mRNA editing enzyme, catalytic polypeptide-like proteins (APOBECs), play critical roles in innate antiviral defense (Samuel, 2011; Sheehy et al, 2003; Wang et al, 2020a). ADARs deaminate adenosines in double strand RNAs (dsRNAs), converting them into inosines (A > I conversion) (Roth et al, 2019; Rusk, 2019). Three ADAR genes have been identified in the human genome, with ADAR1 and ADAR2 widely expressed and catalytically active, while ADAR3, primarily found in the brain, lacks deamination activity (Roth et al, 2019; Rusk, 2019). ADAR1 and ADAR2 exert antiviral effects by destabilizing dsRNA through the introduction of multiple A > I substitutions, a mechanism observed in various RNA viruses, including influenza virus, hepatitis delta virus, lymphocytic choriomeningitis virus, rift valley fever virus, hepatitis C virus and Zika virus (Jayan and Casey, 2002; Piontkivska et al, 2017; Samuel, 2011). The APOBEC family consists of cytosine deaminases that deaminate cytosines and convert them into uracils (C > U) in

[1]Zhejiang Provincial Key Laboratory of Pancreatic Disease, The First Affiliated Hospital, and Institute of Translational Medicine, Zhejiang University School of Medicine, Hangzhou, China. [2]Department of Urology & Andrology, Sir Run Run Shaw Hospital, Zhejiang University School of Medicine, Hangzhou, China. [3]School of Pharmacy, Shanghai University of Traditional Chinese Medicine, Shanghai, China. [4]The Affiliated Hospital of Jiangxi University of Traditional Chinese Medicine, Nanchang, China. [5]Center for Infectious Disease Research, School of Medicine, Tsinghua University, Beijing, China. [6]School of Life Science and Technology, ShanghaiTech University, Shanghai, China. [7]Shanghai Institute for Advanced Immunochemical Studies, ShanghaiTech University, Shanghai, China. [8]Shanghai Key Laboratory of Orthopedic Implants, Department of Orthopedic Surgery, Shanghai Ninth People's Hospital, Shanghai Jiao Tong University School of Medicine, Shanghai, China. [9]Guangzhou Laboratory, Guangzhou International Bio Island, Guangzhou, Guangdong, China. [10]Shanghai-MOST Key Laboratory of Health and Disease Genomics, NHC Key Lab of Reproduction Regulation, Shanghai Institute for Biomedical and Pharmaceutical Technologies, Shanghai, China. [11]These authors contributed equally: Zhean Li, Lingling Luo, Xiaohui Ju, Shisheng Huang. ✉E-mail: chuang@shutcm.edu.cn; huangxx@shanghaitech.edu.cn; qding@tsinghua.edu.cn; zhangy@shanghaitech.edu.cn

single-stranded DNA or RNA. This family comprises eleven members: APOBEC (A)1, A2, A3A, A3B, A3C, A3D, A3F, A3G, A3H, A4 and activation-induced cytidine deaminase (AID) (Salter et al, 2016). APOBEC proteins can restrict the replication of viruses such as HIV and HBV (Stavrou and Ross, 2015).

Analysis of existing SARS-CoV-2 mutations in COVID-19 patients has revealed predominant mutational patterns characterized by A > I (G) and C > U (T) transitions, suggesting that RNA editing mediated by ADARs and APOBECs likely plays a crucial role in SARS-CoV-2 genome mutagenesis, which underlies both antiviral immunity and viral evolution (Di Giorgio et al, 2020; Picardi et al, 2022; Ringlander et al, 2022; Simmonds, 2020; Wang et al, 2020a). Here, we demonstrate that this is indeed the case, and further describe how the virus promotes this mutagenesis, which has therapeutic implications for coronavirus infection. Host deaminases specifically interact with the nucleocapsid (N) protein, a pivotal viral RNA-binding protein in the viral life cycle, thereby targeting viral RNA for editing. Concurrently, the N protein enters deaminase-rich RNA granules through a phase-separation process. This compartmentalization concentrates RNAs and proteins, enhancing interactions between deaminases and the N protein, as well as with viral RNA, and consequently boosting the efficiency of RNA editing on the viral genome. Notably, the N protein with a phenylalanine at position 17 (F17) harnesses host deaminase-associated RNA granules to target the viral genome, potentially modulating the mutagenesis of other coronaviruses. Moreover, a mutant N protein with an alanine substitution at position 17 ($N^{F17A}$) loses the ability to enter stress granules (SGs), which impairs its interaction with deaminases and reduces the SARS-CoV-2 RNA mutagenesis, further supporting the role of deaminase-enriched condensates in viral RNA editing. Our study shed new light on insights into the coronavirus mutagenesis.

# Results

## Deaminase-catalyzed viral RNA editing blocks viral replication while potentially promoting viral mutagenesis

To ascertain the roles of APOBECs and ADARs in SARS-CoV-2 mutagenesis, we utilized a trans-complementation viral infection model in Caco-2 cells using the viral nucleocapsid (N) protein (Ju et al, 2021). With the reference sequence of the SARS-CoV-2 genome (NC_045512), we used RNA-seq to profile the proportion of 12 single nucleotides variant (SNV) types, categorized by the four nucleotide changes (A: A > C, A > G, A > T; C: C > A, C > G, C > T; G: G > A, G > C, G > T; T: T > A, T > C, T > G). Our findings align with the reported mutational patterns (Di Giorgio et al, 2020; Wang et al, 2020a), where APOBECs-driven C > U (C > T/G > A) and ADARs-mediated A > I (A > G/T > C) transitions were the most prevalent (Fig. 1A; Appendix Fig. S1A,B; Dataset EV1). This suggests that host deaminases likely engage in RNA editing on viral genomes and positive/negative-strand transcripts during the viral infection process. However, SARS-CoV-2 infection did not alter the mRNA expression of APOBECs and ADARs (Appendix Fig. S1C), indicating that the observed viral mutations were not due to changes in deaminase expression levels. The secondary mutational groups, G > U (G > T) and C > A transversions, predominant for nucleotides C and G (Fig. 1A; Appendix Fig. S1A), are likely influenced by ROS activity (Valyi-Nagy and Dermody, 2005).

APOBECs- or ADARs-catalyzed deamination is impacted by sequence contexts (Roth et al, 2019; Salter et al, 2016). Indeed, within the SARS-CoV-2 genome, AA and GA motifs are the most and the least preferable, respectively, for A > I mutations (Fig. 1B), whereas A/UCA/U is the preferred motif for C > U transitions (Fig. 1B), consistent with the known substrate preference of ADARs and APOBECs (Bass, 2002; Eggington et al, 2011; Rosenberg et al, 2011; Sharma et al, 2015). These data implicate host deaminases in SARS-CoV-2 RNA genome editing. Notably, each nucleotide variant is evenly represented with respect to SNV frequency, suggesting that the various forces driving SARS-CoV-2 mutation exert relatively equal influence (Appendix Fig. S1D).

To further characterize deaminase-induced SARS-CoV-2 mutation, we performed a series of viral infections in Caco-2 cells and sequenced the viral RNA after the first and sixth passage (P1 and P6). C > U and A > I substitutions were extensive at the first passage suggested that host deaminases induced viral genome mutation, a possible outcome of which is to restrict the viral RNA replication by mutating the viral RNA in the early stages of SARS-CoV-2 infection (Fig. 1C; Dataset EV2). Strikingly, the number of SNVs dramatically decreased at the sixth passage (Fig. 1C; Appendix Fig. S1E,F), concomitant with enrichment of certain mutations such as T2578C (1.5-fold increase in mutation rate), C5219T (6.6-fold increase), A19736G (425.3-fold increase) and C27970T (3.3-fold increase), which do not impair viral stability could be maintained in the subsequent passages (Appendix Fig. S1G, blue box). The data suggest that host-induced mutations typically restrict viral replication, leading to the eventual selection of mutations that are neutral or beneficial for viral adaptation and fitness.

## N protein binds deaminases to promote viral RNA mutation

We next investigated whether SARS-CoV-2 virus hijacks host deaminases to actively promote the mutagenesis. It is known that viral N protein, responsible for assembling the viral RNA genome to form a ribonucleoprotein (RNP) complex, engages host RNA-binding proteins to overcome the host antiviral response (Gordon et al, 2020; Luo et al, 2021). To determine whether the N protein also binds the deaminases, which are by definition RNA-binding proteins (Bishop et al, 2004; Rusk, 2019). GFP-tagged N protein was co-expressed in HeLa cells with Flag-tagged APOBECs (A1, AID, A2, A3 (A to D, E to H), and A4) or ADARs (ADAR1 and ADAR2) before co-immunoprecipitation (Co-IP) analysis. The N protein indeed interacted with ADARs (ADAR1 and ADAR2) and some APOBECs (A1, AID, A3D, A3F, A3G, A3H, but not A2, A3A, A3B, A3C and A4, Appendix Fig. S2A). The N protein also co-immunoprecipitated with endogenous A3G and ADAR2, as expected (Fig. 1D). The N protein-deaminase interaction was RNA-independent, as it persisted with RNase treatment (Appendix Fig. S2B,C). Finally, in HeLa cells expressing the mRNA encoding the N protein (hereafter referred to as "N mRNA"), the anti-ADAR2 antibody pulled down the N mRNA, demonstrating that the N protein, deaminase and the viral RNA formed a ternary complex (Fig. 1E). Together, these data define a novel role of N protein in directing host deaminases to target viral RNA. The SARS-CoV-2 genome encodes some non-structural proteins (nsp) such as nsp7, nsp8, nsp9, nsp12 and nsp16, which aid in viral RNA processing and replication (Gao et al, 2020; Yin et al, 2020). Unlike the N protein, these nsp proteins, as well as structural proteins like spike

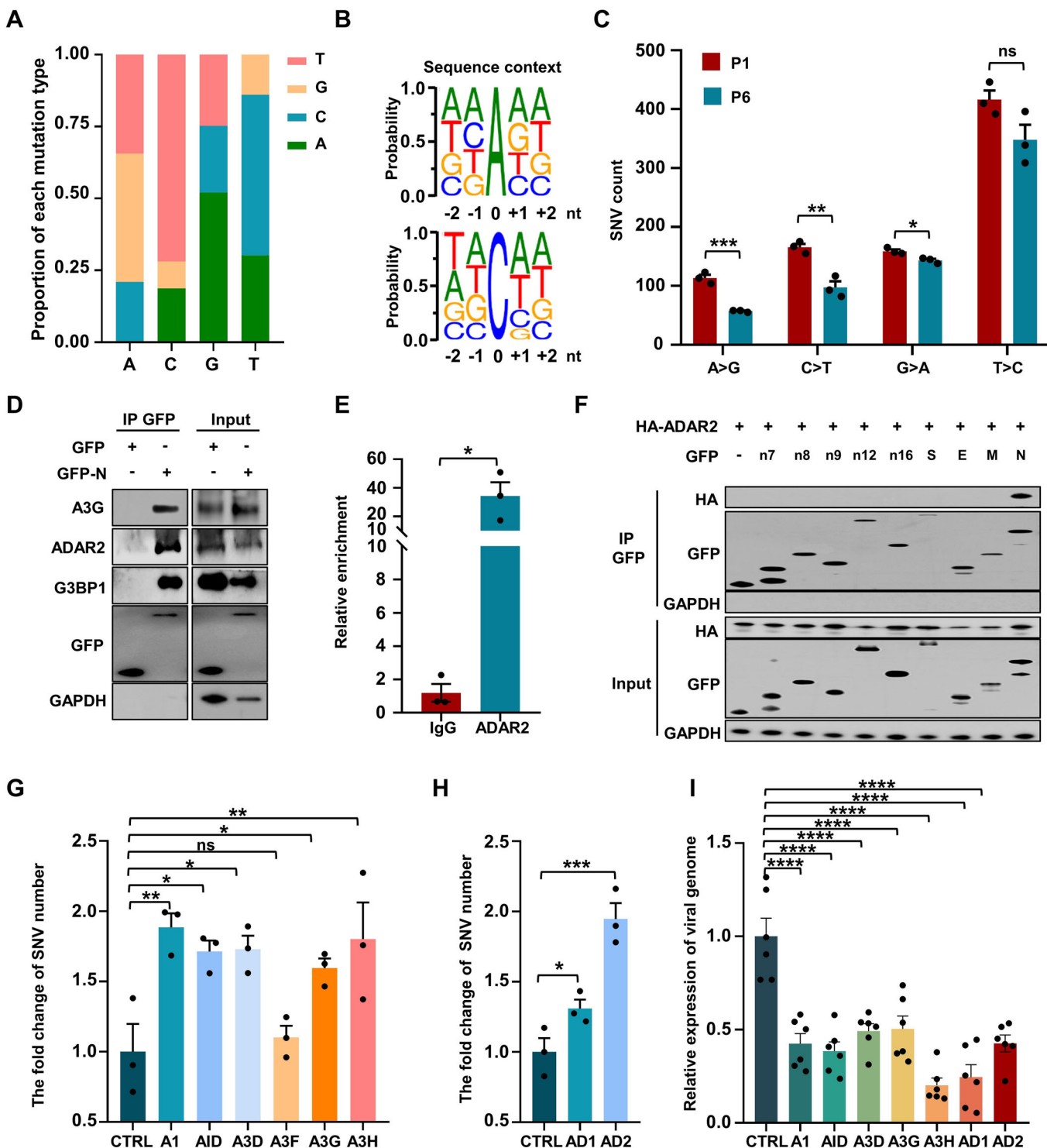

(S) protein, envelope (E) protein and membrane (M) protein, do not interact with the deaminase tested (A3G or ADAR2), demonstrating the uniqueness of the N protein (Fig. 1F; Appendix Fig. S2D). We subsequently explored the functional consequences of N protein-deaminase interactions through two experiments. First, we determined whether N protein interaction with deaminases facilitated editing of its own mRNA. To this end, we transfected the N mRNA into HeLa cells

and analyzed it by sequencing. Predominant mutations were A > I and C > U transitions (Appendix Fig. S3A,B; Dataset EV3), with CCG and C/AAA/G being the least and most favored motifs for C > U and A > I transitions, respectively (Appendix Fig. S3C), aligning with the sequence context of the N gene in the SARS-CoV-2 genome (Appendix Fig. S3D). Of note, the N mRNA also carried non-classic edits such as A > T, G > A and T > C (Li et al, 2011; Tao et al, 2021),

**Figure 1. Deaminases link N protein to edit viral RNA and impair viral production.**

(A) Average proportion of nucleotide changes in SARS-CoV-2 transcriptomes (allelic fraction ≥0.1%). Mean with SE were plotted from three independent biological replicates. The x-axis represents the original nucleotide of the SARS-CoV-2 genome, and indicative colour blocks represent nucleotides to which original nucleotide mutated. (B) Sequence contexts for A > I (G) and C > U (T) mutations in the viral transcriptome. (C) The number of SNVs identified in the SARS-CoV-2 transcriptomes following one (P1) and six passages (P6, allelic fraction ≥0.1%). Mean with SE were plotted from three biological replicates. Statistical analysis was performed with a two-tailed unpaired t-test. ns >0.05, *P < 0.05, **P < 0.01, ***P < 0.001. ns, not significant. (D) The SARS-CoV-2 N protein interacts with host endogenous G3BP1, A3G and ADAR2. HeLa cells were transfected with plasmids encoding GFP-tagged N protein, followed by immunoprecipitation with an anti-GFP antibody and the bound proteins were then analyzed by western blotting. (E) Enrichment of N protein mRNA, as measured by RIP assay in HeLa cells. Cells were transfected with the N gene with a GFP tag and lysates were collected at 48 h post-transfection. Lysates were divided for incubation with a mouse anti-ADAR2 antibody, and co-precipitated RNA was analyzed by RT-qPCR. Mean with SE were plotted from three biological replicates. Statistical analysis was performed with a two-tailed unpaired t-test. *P < 0.05. (F) ADAR2 specifically interacts with the SARS-CoV-2 N protein. HeLa cells were co-transfected with plasmids encoding HA-tagged ADAR2 and GFP control or GFP-tagged SARS-CoV-2 genes, including nsp7 (n7), nsp8 (n8), nsp9 (n9), nsp12 (n12), nsp16 (n16), spike protein (S), envelope protein (E), membrane protein (M) and nucleocapsid protein (N). (G) APOBECs mediate SARS-CoV-2 RNA mutagenesis (allelic fraction ≥2%). Mean with SE were plotted from three biological replicates. The number of SNVs with C > T/G > A mutations was normalized to the number of control group. Statistical analysis was performed with a one-way ANOVA test. ns > 0.05, *P < 0.05, **P < 0.01. ns, not significant. (H) ADARs mediate SARS-CoV-2 RNA mutagenesis (allelic fraction ≥2%). Mean with SE were plotted from three biological replicates. The number of SNVs with A > G/T > C mutations was normalized to the number of control group. Statistical analysis was performed with an unpaired t-test. *P > 0.05, ***P < 0.001. (I) Deaminases regulate SARS-CoV-2 production. Viral genomic RNA expression was normalized to that of the control group. Mean with SE were plotted from six biological replicates. Statistical analysis was performed with a one-way ANOVA test. ****P < 0.0001. Source data are available online for this figure.

revealing additional RNA mutagenesis mechanisms. Thus, the specific interaction between the N protein and host deaminases is a potential mechanism for host-directed A > I and C > U RNA editing on the SARS-CoV-2 genome. Second, we assessed whether the ability of deaminases to bind the N protein correlated with their capacity to edit the SARS-CoV-2 genome. To this aim, we constructed stable Caco-2 cell lines expressing various APOBECs and ADARs, and infected these cells with SARS-CoV-2 virus using an N protein-based trans-complementation model (Appendix Fig. S4A). Remarkably, among the six APOBECs capable of binding the N protein (A1, AID, A3D, A3G, A3H and A3F), all but A3F efficiently induced C > U mutations at various sites (Fig. 1G; Appendix Fig. S4B), some of which were also observed in Delta and Omicron subclinical groups (Appendix Fig. S4C) (Saifi et al, 2022). In contrast, none of the APOBECs lacking N protein interaction (A3A, A3B and A4) could boost viral genome mutagenesis. A3F showed negligible editing activity on the SARS-CoV-2 genome, probably due to a lack of viral RNA editing activity (Appendix Fig. S4E). Similarly, ADAR1 and, particularly, ADAR2 increased A-to-I mutations (Fig. 1H; Appendix Fig. S4D). Deaminases exhibit a remarkable innate antiviral defense by introducing mutations into the viral genome (Samuel, 2011; Turelli et al, 2004). As expected, deaminases capable of mutating the viral genome, namely APOBECs (A1, AID, A3D, A3G, A3H) and ADAR1/2, all decreased SARS-CoV-2 production and infection (Fig. 1I). We conclude that the viral N protein binds the deaminases to promote viral genome mutation.

## Co-localization of N protein, deaminase and viral RNA to SGs enhances viral RNA mutagenesis

SGs are cytoplasmic membrane-less dynamic structures containing mRNA-protein aggregates and formed in response to viral infection, including SARS-CoV-2 infection. They can sequester viral mRNA to inhibit viral replication (Buchan and Parker, 2009; Lloyd, 2013; White and Lloyd, 2012; Yang et al, 2020; Zheng et al, 2021). To test whether host deaminases co-localized with the N protein to SGs, we first co-expressed GFP-tagged N protein and HA-tagged A3G or ADAR2 in HeLa cells and analyzed their association using immunoprecipitation. The N protein interacted with host G3BP1 and A3G or ADAR2 (Fig. 2A), suggesting that G3BP1 and host deaminases act as host binding partners of the

viral N protein and may be jointly involved in SGs. Subsequently, Cells transfected with plasmids encoding mCherry-tagged N protein and BFP-tagged APOBECs (AID, A3G and A3H) were immunostained with an antibody against endogenous G3BP1 (a SG marker). Under arsenite (AS)-induced stress condition, N protein specifically co-localized with G3BP1 and AID, A3G or A3H to SGs, but not other SARS-CoV-2 proteins (Figs. 2B and EV1A,B; Appendix Fig. S5A). ADAR1p110 is primarily expressed in the nucleus, while ADAR1p150 can move between the nucleus and cytoplasm as a shuttling protein (Shiromoto et al, 2021). ADAR2 is predominantly nuclear, but a small fraction is cytoplasmic (Aizawa et al, 2010; Behm et al, 2017; Jimeno et al, 2021; Marcucci et al, 2011). We confirmed the reported subcellular localization of ADAR1 and ADAR2 (Fig. 2B; Appendix Fig. S5B) (Jimeno et al, 2021; Marcucci et al, 2011; Sakurai et al, 2017; Weissbach and Scadden, 2012). To further confirm the cytoplasmic presence of ADAR2, we performed the subcellular fractionation assays, followed by immunoblotting detection using GAPDH as a cytoplasmic protein marker and Lamin A/C as a nuclear protein marker. ADAR2 primarily localized to the nucleoplasm and nucleoli, with only a small fraction present in the cytoplasm (Appendix Fig. S6A–G) (Jimeno et al, 2021; Marcucci et al, 2011). Accordingly, ADAR1 and ADAR2 partially co-localized with the N protein in AS-induced SGs (Fig. 2B; Appendix Fig. S5B). The co-localization of deaminases with the N protein was also observed in SGs induced by other stressors, including polyI:C (a viral infection mimic), DTT (an ER stressor) and sorbitol (an osmotic stressor) (Fig. EV2A–C), suggesting that the N protein is effectively incorporated into deaminase-localized RNA granules. SGs can concentrate RNAs and RNA-binging proteins, promoting their interactions and accelerating biochemical reactions (Banani et al, 2017; Kent et al, 2020; Tsang et al, 2019). Indeed, in HeLa cells expressing the N protein, AS enhanced the interaction among the deaminases, the N protein and its associated RNA (Fig. 2C,D). Similarly, in HeLa transfected with the N mRNA, SG formation increased the numbers of mutations in the mRNA (Fig. 2E,F; Dataset EV4). Notably, the mutation signature remained unaffected, indicating that SG formation accelerated deaminase reaction without altering the intrinsic property of the enzymes, as expected (Appendix Fig. S7A–C; Dataset EV4). These data suggest that the

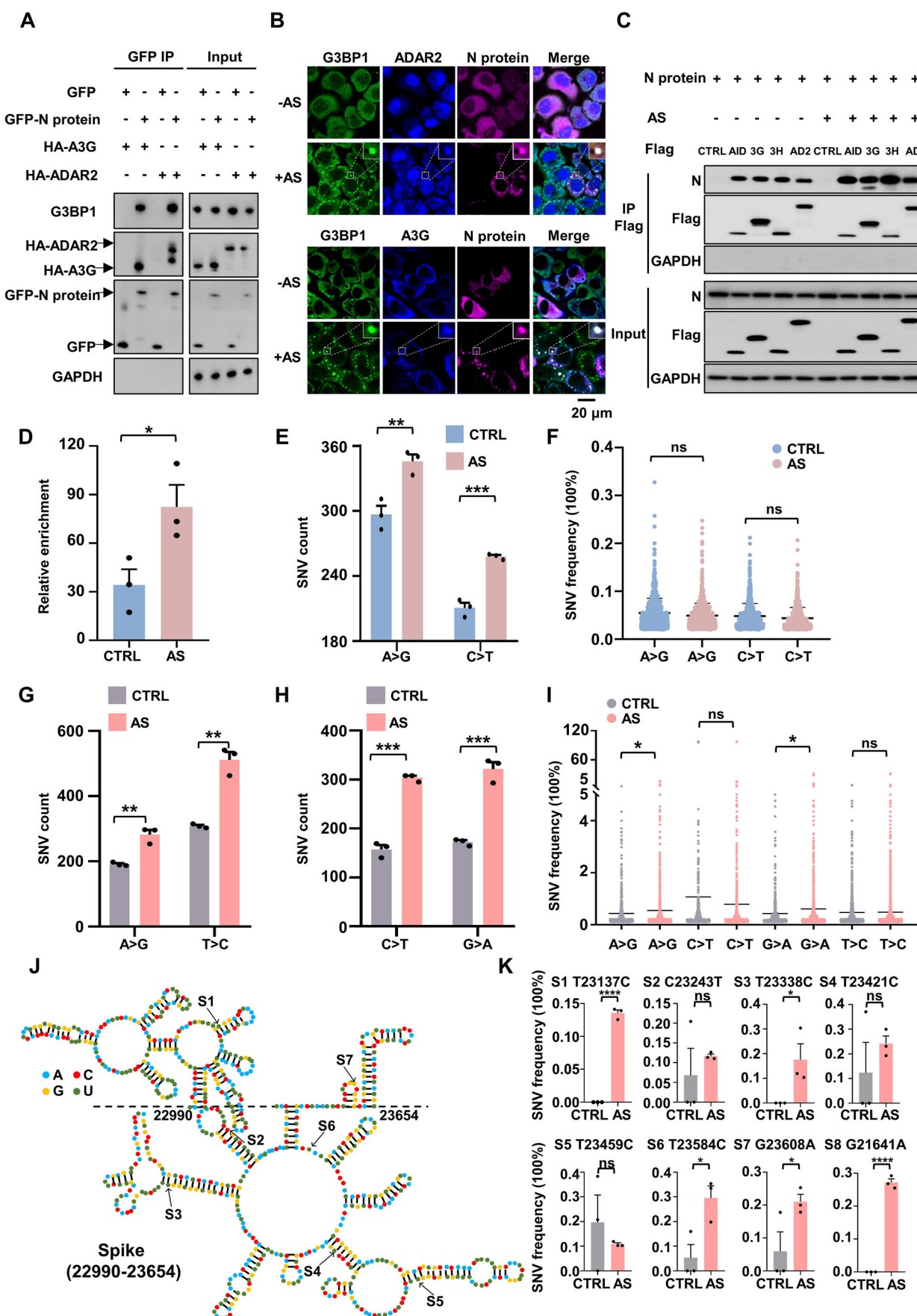

◀ **Figure 2.  SARS-CoV-2 N protein and viral RNA enter deaminase-enriched condensates to efficiently promote viral RNA mutagenesis.**

(A) The SARS-CoV-2 N protein interacts with host G3BP1 and deaminase A3G or ADAR2. HeLa cells were co-transfected with plasmids encoding GFP-tagged N protein or GFP control and HA-tagged A3G or ADAR2. Cell lysates were immunoprecipitated with an anti-GFP antibody, and the precipitated proteins were analyzed by western blotting with antibodies against HA or G3BP1. (B) A3G or ADAR2 co-localizes with the N protein in SGs in HeLa cells. HeLa cells transfected with the N gene and A3G or ADAR2 were treated with 200 μM AS for 45 min to induce SG formation, followed by immunofluorescence staining for the N protein, A3G or ADAR2 and endogenous G3BP1. Scale bar: 20 μm. (C) Condensate formation of N protein-deaminases complex enhances the interaction between the N protein and deaminases. HeLa cells transfected with the N gene and Flag control or Flag-tagged deaminases, including AID, A3G (3G), A3H (3H) and ADAR2 (AD2), were treated with or without AS. (D) mRNA enrichment of the N protein was measured by RIP assay in HeLa cells treated with or without 200 μM AS treatment. Mean with SE were plotted from three biological replicates. Statistical analysis was performed with a two-tailed unpaired t-test. *$P < 0.05$. (E) Condensate formation promotes the count of SNVs (A > G and C > T mutations) identified in the N protein mRNA in HeLa cells. Values and error bars were represented as the mean ± SEM of three independent biological replicates (allelic fraction ≥0.02%). Statistical analysis was performed with a two-tailed unpaired t-test. **$P < 0.01$, ***$P < 0.001$. (F) Allelic fraction of SNVs (A > G and C > T mutations) in the N gene in HeLa cells with or without AS treatment. Mean with SE were plotted from three biological replicates. Statistical analysis was performed with a two-tailed unpaired t-test. ns > 0.05. ns, not significant. (G, H) Formation of N protein-deaminase complex-containing RNA condensates increases the number of SNVs identified in the SARS-CoV-2 genome. (G) A > G and T > C mutations; (H) C > T and G > A mutations. Values and error bars were represented as the mean ± SEM of three independent biological replicates (allelic fraction ≥0.1%). Statistical analysis was performed with a two-tailed unpaired t-test. **$P < 0.01$, ***$P < 0.001$. (I) Allelic fraction of the indicated SNVs in the SARS-CoV-2 genome in infected cells with or without AS. Mean with SE were plotted from three biological replicates. Statistical analysis was performed with a two-tailed unpaired t-test. ns >0.05, *$P < 0.05$. ns, not significant. (J) Representative secondary structure prediction of the region (22,990-23,654nt) extracted from the spike gene based on icSHAPE data (Sun et al, 2021a). The edited base is indicated by arrows. Site, S. (K) Condensate formation of the N protein-deaminases complex promotes the mutational rate of indicated sites in the spike gene. Mean with SE were plotted from three biological replicates. Statistical analysis was performed with an unpaired t-test. ns > 0.05, *$P < 0.05$, ****$P < 0.0001$. ns, not significant. Source data are available online for this figure.

condensation potentially enhanced the viral RNA-deaminase interaction, thereby promoting viral RNA mutagenesis.

To elucidate the role of such condensates in viral RNA mutagenesis, we treated infected cells with AS to induce the formation of RNA granules, after which RNA was harvested and subjected to sequencing analysis. The N protein is recruited into deaminase-enriched SGs (Appendix Fig. S7D,E), resulting in a significant increase in the number of A > G/T > C and C > T/G > A substitutions (Fig. 2G,H; Appendix Fig. S7F,G; Dataset EV5). Notably, such condensed compartmentalization greatly promoted the frequency of A > G/T > C and C > T/G > A transitions across the viral genome, particularly leading to the emergence of high-frequency mutations exceeding 5% (Fig. 2I). The spike gene is a principal determinant for the SARS-CoV-2 transmission and resistance to antibody neutralization (Liu et al, 2021). In alignment with the whole-genome findings, the formation of deaminase-enriched RNA granules resulted in a 1.23-fold increase in A > G/T > C mutations and a 1.54-fold increase in C > T/G > A mutations (Appendix Fig. S7H). The Omicron variant is characterized by approximately 32 mutations within the spike protein, predominantly situated in the N-terminal domain (NTD, pivotal for antibody recognition) and the receptor-binding domain (RBD, responsible for ACE2 binding to facilitate viral entry) (Chi et al, 2020; Planas et al, 2022). The number of SNVs was significantly elevated, with a 3.7-fold increase in A > G/T > C mutations and a 3.5-fold increase in C > T/G > A mutations, when the N protein and viral RNA were concentrated within deaminase-enriched RNA granules, despite the overall SNV frequency remaining relatively low (Appendix Fig. S7I,J). Indeed, RNA secondary structures, characterized by features such as mismatches, bulges, stem loops, terminal loops or internal loops have been implicated in modulating the efficiency and specificity of deaminase-mediated editing (Maris et al, 2005; Richardson et al, 1998; Wang et al, 2018). Additionally, the N protein has a preference for binding long, structured RNA (Dinesh et al, 2020). The structural characterization of SARS-CoV-2 RNA reveals a network of well-folded regions within the RNA of the spike protein including many stem loops, internal loops and bulges (Sun et al, 2021b). Within a specific

region (22990nt-23654nt), condensate assembly, which concentrates RNAs and deaminases, greatly improved editing probability and efficiency (S3 and S4 at mismatch, S6 at multibranch loop and S7 at hairpin loop; Fig. 2J,K). Interestingly, SG formation triggered RNA mutagenesis at nucleotide G21641 within the alanine residue at position 27 of the spike protein (S8; Fig. 2K), a site present in Omicron variants of concern (VOC), strongly suggesting that C > U and A > I mutations induced by APOBECs and ADARs may enhance SARS-CoV-2 fitness. These results demonstrate that the N protein and viral RNA leverage deaminase-enriched granules to promote SARS-CoV-2 genome mutagenesis and evolution.

G3BP1/2 serve as core proteins in the assembly of SGs, and the cells deficient in G3BPs fail to form SGs in response to AS treatment (Yang et al, 2020). Specifically, we used CRISPR/Cas9 to generate HeLa cells with the double knockout of G3BP1 and G3BP2 (referred to as G3BP1/2 dKO; Fig. EV3A), and then tested the ability of SG assembly to enhance editing at transfected N mRNA. We confirmed that G3BP1/2 dKO cells indeed failed to form deaminase-containing SGs after AS treatment (Fig. EV3B–D), thereby disrupting the condensate formation of N protein-deaminases (Fig. 3A). To validate the role of G3BP1/2 in recruiting N protein into deaminase-enriched condensates, we treated G3BP1/2 dKO cells with sorbitol, a hyperosmotic stressor that triggers the G3BP-independent SG-like foci formation. Interestingly, the absence of G3BP1/2 significantly reduced, but did not completely abolish, the entry of N protein into SG-like foci under hyperosmotic stress (Fig. 3B,C), indicating that G3BP1/2 is required for the recruitment of N protein into G3BP1/2-dependent SGs, rather than G3BP1/2-independent SGs. Furthermore, the interaction with G3BP1/2 contributes to the recruitment of more N proteins and RNAs into SGs. We then validated the impact of condensate assembly on the deaminase activity. Notably, G3BP1/2 deficiency did not significantly affect the APOBECs and ADARs-mediated RNA-editing capability compared to WT cells (Appendix Fig. S8A–C; Dataset EV6). Conversely, G3BP1/2 deficiency, which disrupted the formation of N protein-deaminase condensates, markedly impaired the APOBECs and ADARs-mediated RNA-editing efficiency and failed to increase the number of SNVs

**A**

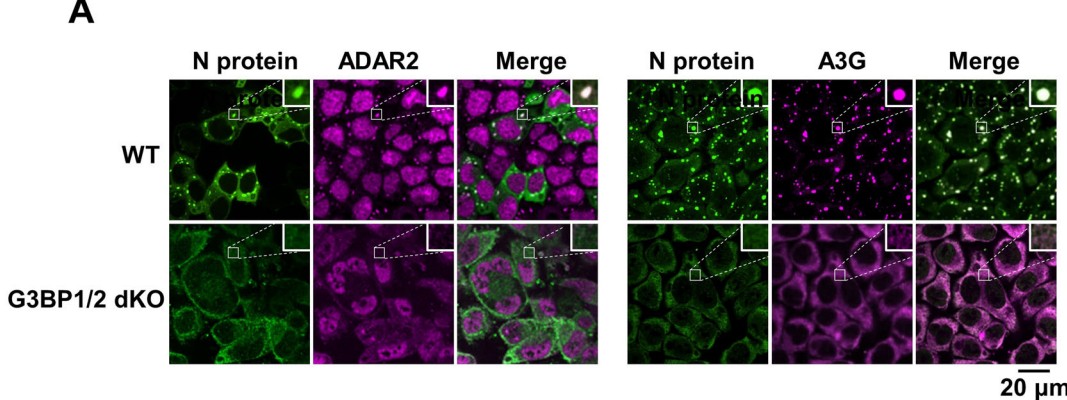

**B**

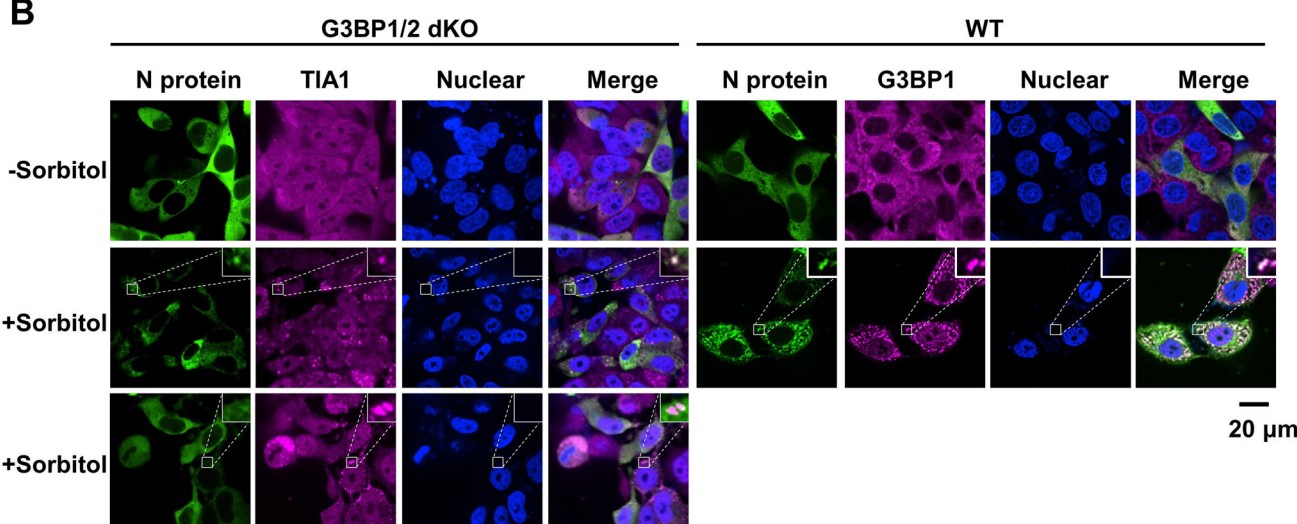

**C**                          **D**

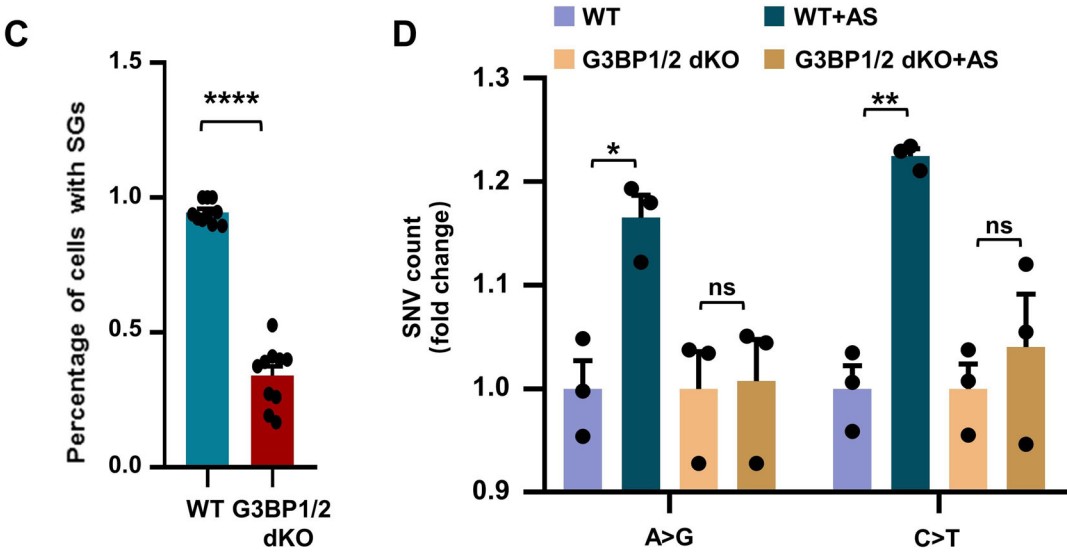

**Figure 3.  Disruption of N protein/deaminase-containing condensates attenuates viral RNA mutagenesis.**

(A) Deletion of G3BP1/2 (G3BP1/2 dKO) in HeLa cells abrogates the formation of N protein/deaminase-containing condensates in G3BP1/2-null HeLa cells. Scale bar: 20 μm.
(B) Immunofluorescence analysis showing the localization of the N protein in SGs in cells with or without G3BP1/2 under sorbitol-induced stress. The WT or G3BP1/2-depleted HeLa cells transfected with the N gene were treated with 0.5 M sorbitol for 1 h to induce SG formation, followed by immunostaining for N protein, and endogenous G3BP1 or TIA1. Scale bar: 20 μm.
(C) Quantification of SGs, expressed as the percentage of cells containing SGs in fixed WT or G3BP1/2 dKO HeLa cells with or without sorbitol treatment. Mean with SE were plotted from ten biological replicates. Statistical analysis was performed with an unpaired t-test. ****$P < 0.0001$. (D) Disruption of N protein/deaminase-containing condensates attenuates ADARs and APOBECs-mediated RNA editing in the N protein mRNA. Values and error bars were represented as the mean ± SEM of three independent biological replicates (allelic fraction ≥0.02%). Statistical analysis was performed with a one-way ANOVA test. ns > 0.05, *$P < 0.05$, **$P < 0.01$. ns, not significant. Source data are available online for this figure.

(Fig. 3D; Appendix Fig. S8D,E; Dataset EV7). Thus, these results further reveal that N protein and viral RNA enter deaminase-localized granules, thereby facilitating viral RNA mutagenesis.

## The F17A mutation in N protein disrupts localization to deaminase-enriched SGs and impairs viral RNA mutagenesis

The F17A mutation within intrinsically disordered region 1 (IDR1) of the N protein has been shown to markedly disrupt the N-G3BP1 interaction (Huang et al, 2021; Yang et al, 2024), suggesting that $N^{F17A}$ variant may be incapable of entering deaminase-enriched SGs. We generated a GFP-tagged $N^{F17A}$ construct and conducted Co-IP assays in HeLa cells (Fig. 4A). In line with previous GST pull-down assay findings (Huang et al, 2021), the F17A substitution significantly impaired the N protein-G3BP1 interaction (Appendix Fig. S9A), which in turn substantially reduced the N protein's ability to phase separate with G3BP1 and localize to SGs (Fig. 4A–C; Appendix Fig. S9B,C). However, the F17A mutation retained the ability to bind ADAR2/A3G (Fig. 4D; Appendix Fig. S9D), suggesting that residue 17 F is a specific motif of the N protein to interact with G3BP1, rather than deaminases. As a consequence, the mutant failed to bring the viral RNA into deaminase-accumulated SGs (Fig. 4E), thus weakening the interaction between ADAR2 and the N protein RNA due to the loss of their spatial proximity (Fig. 4F). We then investigated whether preventing N protein and its bound RNA from entering deaminase-containing RNA granules affects deaminases-mediated RNA editing activity on the N protein RNA. To this end, we transfected the $N^{WT}$ or $N^{F17A}$ protein mRNA into HeLa cells and then induced SG formation. Notably, the F17A substitution did not eliminate the deaminase-mediated RNA editing level for the $N^{F17A}$ gene, as the $N^{F17A}$ protein retains the ability to interact with deaminases, thereby targeting its RNA (Appendix Fig. S9E,F; Dataset EV8). However, the reduced RNA enrichment of the N gene in ADAR2 failed to enhance deaminase activity (Fig. 4F,G; Appendix Fig. S9G,H; Dataset EV8), further highlighting the role of deaminase-containing RNA granules in viral RNA mutagenesis. Similar effects were observed at the viral genome in an $N^{F17A}$-based transcomplementation SARS-CoV-2 cell culture model. The F17A mutation did not change the mutational signature that APOBECs-mediated C > U (C > T/G > A) and ADARs-mediated A > I (A > G/T > C) transitions, which remained the predominant mutational types (Appendix Fig. S10A,B; Dataset EV9). Importantly, the F17A mutation led to the exclusion of N protein and its associated RNA from deaminase-containing RNA granules. This exclusion impaired the enhancement of deaminase activity on viral RNA, consequently resulting in

a reduction in the proration of C > U and A > I mutations (Fig. 4H; Appendix Fig. S10A,B; Dataset EV9). Furthermore, the $N^{F17A}$ protein lacked the ability to co-localize with deaminase in cytoplasmic foci, thereby failing to promote deaminase activity and achieve a high frequency of SNVs (C900T, A4870G, C20270T and G29528A) compared to results based on WT N protein (Fig. 4I,J; Appendix Fig. S10C–F). We conclude that the N protein drives viral RNA into deaminase-containing SGs to facilitate viral RNA mutagenesis.

## The RNA binding domain is essential for ADAR2 to form RNA-protein condensates with the N protein

ADAR1/2 comprise two or more double-stranded RNA binding domains (dsRBDs) and a C-terminal adenosine-deaminase domain (Rusk, 2019). To identify the individual domains of ADARs that contribute to their interaction with the N protein and condensate formation, we selected ADAR2 and generated a series of truncated mutants, including dsRBD1 (RNA binding domain 1, aa 78–142), dsRBD2 (RNA binding domain 2, aa 230–296), deaminase domain (aa 317–710), ΔdsRBD1 (lack of dsRBD1), ΔdsRBD2 (lack of dsRBD2) and ΔDea (lack of deaminase domain) (Fig. 5A). We found that dsRBD1 alone showed robustly interacted with the N protein (Fig. 5A,B). In contrast, neither dsRBD2 nor the deaminase domain alone was sufficient to form a complex with the N protein. Furthermore, the absence of dsRBD1 completely abolished the ADAR2 interaction with N protein (Fig. 5B). The mutants ΔdsRBD2 and ΔDea, which retain dsRBD1, maintained the ability to interact with the N protein (Fig. 5B), suggesting that dsRBD1 is the domain within ADAR2, responsible for forming a protein complex with the N protein. Next, we evaluated which domains of ADAR2 are involved in the co-localizing with the N protein to SGs. However, ADAR2 lacking dsRBD1 failed to form protein condensates with the N protein (Fig. 5C; Appendix Fig. S11A), demonstrating that dsRBD1 is essential for ADAR2 to form the condensates with the N protein. SG assembly is regulated by phase separation and the N protein phase separated with G3BP1 into SGs (Luo et al, 2021; Yang et al, 2020), prompting us to investigate whether ADAR2 exhibits features of liquid-like condensates. Phase separation was undetectable for recombinant GFP-dsRBD1 fusion protein (Fig. 5D,E). Remarkably, micrometer-sized liquid droplets rapidly formed upon mixing with RNA extracted from HeLa cells, accompanied by increased fluorescence intensity and larger droplet area (Fig. 5D,E). Fluorescence recovery after photobleaching (FRAP) analysis confirmed that GFP-dsRBD1 diffused rapidly within droplets, and exhibited liquid-like property (Fig. 5F). Additionally, the N protein partitioned into dsRBD1-RNA droplets and enhanced phase separation (Fig. 5G; Appendix Fig. S11B).

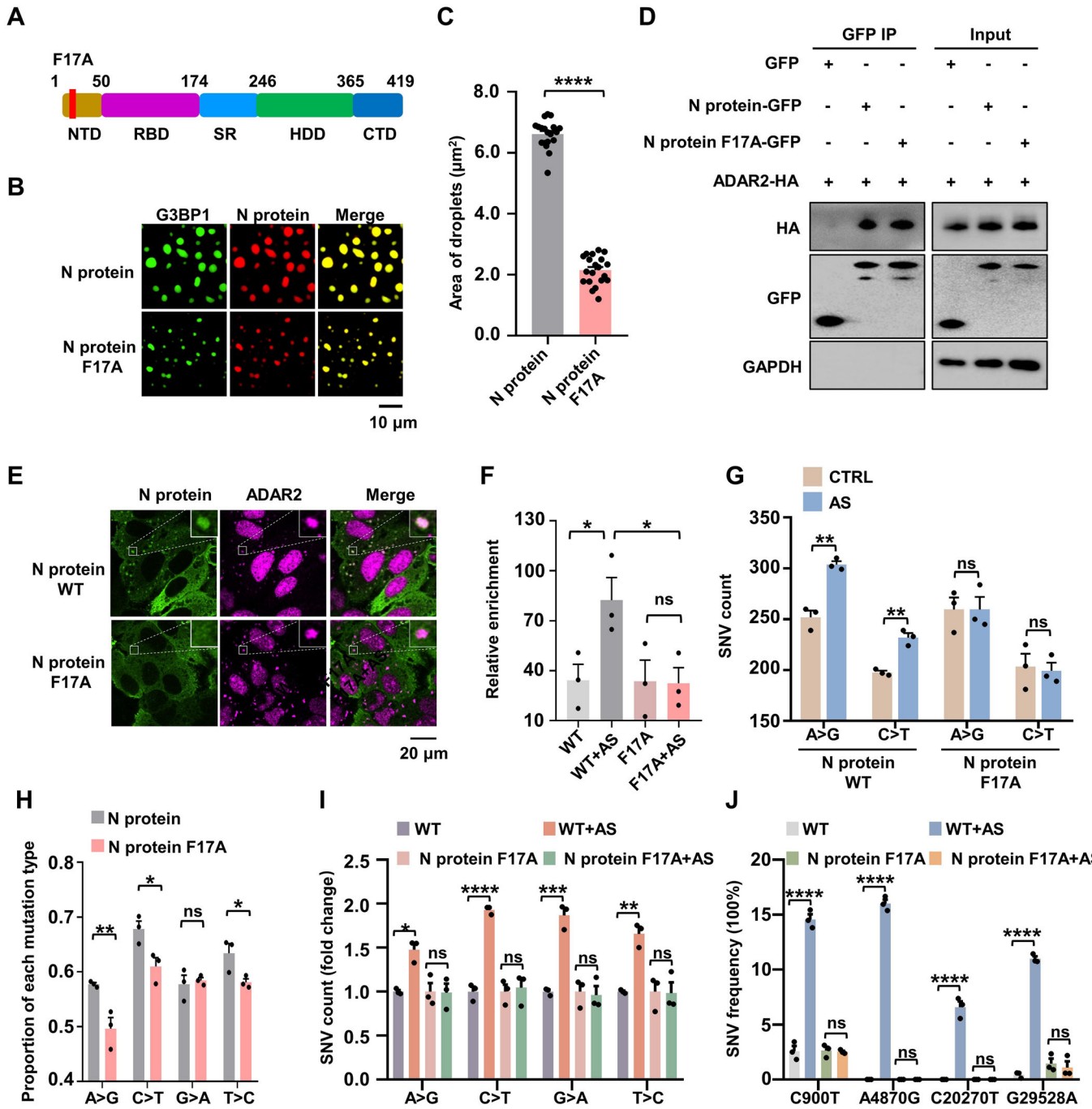

Together, these results revealed that N protein-ADAR2 underwent an
RNA-dependent phase separation to form RNA-protein condensates.

## Conservation of N protein function among various coronavirus

Among the structural proteins of coronaviruses, N proteins, which are responsible for viral genome packaging, exhibit significant sequence conservation. This observation prompted us to investigate the commonalities and differences in the functions of N proteins in the mutagenesis of various coronaviruses. We have shown that SARS-

CoV-2 N protein can undergo phase separation, binds deaminase to promote viral mutations. We now wish to determine whether these functions are conserved among the N proteins from 5 other coronaviruses, namely MERS-CoV, bat-CoV, pangolin-CoV, civet-CoV and SARS-CoV. All N proteins comprise an intrinsically disordered N-terminal domain (N-IDR), an RNA-binding domain (RBD), a Ser/Arg (SR)-rich central disordered region (SR-IDR), a homodimerization domain (HDD) and a disordered C-terminal domain (C-IDR). Disorder prediction analysis reveals that all the N proteins exhibit a high phase separation propensity (Fig. 6A). SARS-CoV-2 N-IDR (residues 1-50) is critical for the N protein phase

**Figure 4. The N^F17A protein fails to enter deaminase-localized RNA condensates, impairing deaminases-mediated editing activity on viral RNA.**

(A) Schematic domain structure of the N^F17A protein. NTD: N-terminal domain; RBD: RNA-binding domain; SR: serine/arginine rich motif; HDD: homodimerization domain; CTD: C-terminal domain. (B) The F17A mutation in the N protein impairs phase separation with G3BP1. Scale bar, 10 μm. (C) Column scatter charts display the droplet area from reactions shown in (B). Data are shown as mean ± SEM (n = 20 independent images). Statistical analysis was performed with an unpaired t-test. ****P < 0.0001. (D) The N^F17A protein retains the ability to interact with ADAR2. HeLa cells were co-transfected with plasmids encoding GFP-tagged N wild type or N^F17A mutant and HA-tagged ADAR2. (E) The N^F17A protein fails to enter deaminase-containing RNA condensates. HeLa cells co-transfected with N^WT or N^F17A mutant were treated with AS for 45 min to induce SGs, followed by immunostaining for N protein (green) and the endogenous ADAR2 (red). (F) Enrichment of the N protein mRNA, as measured by RIP assay in HeLa cells with or without AS treatment. Cells were transfected with N^WT or N^F17A mutant with a GFP tag for 48 h and then treated with or without AS treatment for 45 min. Values and error bars were represented as the mean ± SEM of three independent biological replicates. Statistical analysis was performed with one-way ANOVA. ns > 0.05, *P < 0.05. ns, not significant. (G) The N^F17A protein fails to enter deaminase-containing condensates, impairing deaminases-mediated RNA editing activity on N protein mRNA. Values and error bars were represented as the mean ± SEM of three independent biological replicates (allelic fraction ≥0.02%). Statistical analysis was performed with an unpaired t-test. ns > 0.05, **P < 0.01. ns, not significant. (H) Average proportion of nucleotide changes in the SARS-CoV-2 transcriptomes. Values and error bars were represented as the mean ± SEM of three independent biological replicates. Statistical analysis was performed with unpaired t-test. ns >0.05, *P < 0.05, **P < 0.01. ns, not significant. (I) Fold change in the number of SNVs in the SARS-CoV-2 transcriptomes of infected cells with or without AS treatment. Values and error bars were represented as the mean ± SEM of three independent biological replicates. Statistical analysis was performed with a one-way ANOVA test. ns > 0.05, *P < 0.05, **P < 0.01, ***P < 0.001, ****P < 0.0001. ns, not significant. (J) Allelic fraction of SNVs for C900T, A4870G, C20270T and G29528A in the SARS-CoV-2 transcriptomes. Values and error bars were represented as the mean ± SEM of three independent biological replicates. Statistical analysis was performed with a one-way ANOVA test. ns >0.05, ****P < 0.0001. ns not significant. Source data are available online for this figure.

separation, and the residue 17 F within N-IDR is essential for the N protein's association with G3BP1 (Huang et al, 2021; Luo et al, 2021). We then performed a protein sequence alignment to compare N-IDR sequence among SARS-CoV-2, MERS-CoV, bat-CoV, pangolin-CoV, civet-CoV and SARS-CoV. The analysis revealed that residue 17 F is highly conserved in the N-IDR across different coronaviruses (Fig. 6B), suggesting coronavirus N proteins with this residue are likely capable of interacting with the host G3BP1. Consistent with the sequence conservation, other coronavirus N proteins also showed a high level of interaction with host endogenous G3BP1 and the deaminases tested (A3G and ADAR2; Figs. 6C and EV4A), and they enter the condensates of the N-ADAR1/2 complex once SGs are formed (Fig. EV4B,C) in a G3BP1/2-dependent manner (Appendix Fig. S12A). Interestingly, the MERS-CoV N protein efficiently formed condensates with ADAR2 but not A3G (Figs. 6D,E and EV4C,D), perhaps due to its relatively low sequence homology with the SARS-CoV-2 N protein (Krishnamoorthy et al, 2020). Next, we transfected mRNA encoding N proteins from MERS-CoV and SARS-CoV into cells. Subsequently, the cells were treated with a stressor to induce SG formation. In line with the above results of SARS-CoV-2, APOBEC-regulated C > U (C > T) and ADAR-mediated A > I (A > G) transitions were the dominant mutation categories for nucleotides A and C (Appendix Fig. S12B–D; Datasets EV10,11). Furthermore, the condensate assembly enhanced the A > I mutagenesis in the N mRNA of MERS-CoV and SARS-CoV, including both the number and frequency of SNVs (Fig. 6F,G; Appendix Fig. S12B–D). However, the inducer to form RNA-deaminase condensates did not enhance the APOBEC-mediated C > U mutagenesis in the mRNA of MERS-CoV N protein, unlike what was observed with the SARS-CoV N protein (Fig. 6F; Appendix Fig. S12E,F), possibly due to the low ability of MERS-CoV N protein to enter APOBEC-containing SGs. Collectively, these results demonstrate the evolutionary conservation of N protein functions in spurring viral RNA mutagenesis.

## Discussion

Within the dynamics of host-virus interactions, RNA editing mediated by APOBECs and ADARs often serves as a host immune response to counteract viral infection (Olson et al, 2018; Samuel,

2011; Wang et al, 2020a). In turn, viruses have been shown to exploit host deaminases to enhance genetic diversity and facilitate viral evolution (Dolja, 2009; Wang et al, 2020b). In this study, we employed an N protein-based transcomplementation SARS-CoV-2 cell culture model to comprehensively characterize the RNA mutational signature and viral replication dynamics of SARS-CoV-2, providing experimental evidence that RNA editing by host can both enhance viral mutation and disrupt viral replication. Mechanistically, we demonstrated that the SARS-CoV-2 N protein acts as a bridging molecule, guiding host deaminases to target viral RNA for editing. Furthermore, the N protein enters deaminase-enriched RNA granules through an RNA-dependent phase separation process, leading to the formation of N protein-deaminase condensates. This organizational strategy not only concentrates RNAs and deaminase-associated proteins but also enhances the efficiency of deaminases-catalyzed RNA editing on the viral genome. Such host-directed RNA editing may also regulate the mutagenesis of other coronaviruses possessing a similar residue (F17) in their N proteins. Importantly, the SARS-CoV-2 N protein^F17A fails to enter deaminase-enriched RNA granules, greatly impairing host deaminases activity on viral RNA mutagenesis. Our results thus underscore a mechanism involving phase separation-regulated N protein-deaminase condensation that enables deaminases to drive viral mutation and evolution.

RNA virus mutations often arise from spontaneous errors during viral replication and host-driven antiviral defense systems. Coronaviruses, such as SARS-CoV-2, possess proof-reading machinery in their replication processes, achieved by the nsp14 exonuclease, which confers relatively high fidelity in viral transcription and replication (Gordon et al, 2020). Despite this high-fidelity replication system, the continuous emergence of new mutations in the SARS-CoV-2 genome implies the involvement of host-driven forces in viral genome mutation. Interestingly, the dominant mutational pattern mediated by deaminases has also been displayed in other coronaviruses, including MERS-CoV and SARS-CoV, during their spread (Krishnamoorthy et al, 2020; Yi et al, 2021). In this study, we examined the SARS-CoV-2 RNA sequence in cell culture and found the A > I (G) and C > U (T) mutations are the predominant mutational patterns, consistent with database analysis of SARS-CoV-2 genomic variations and

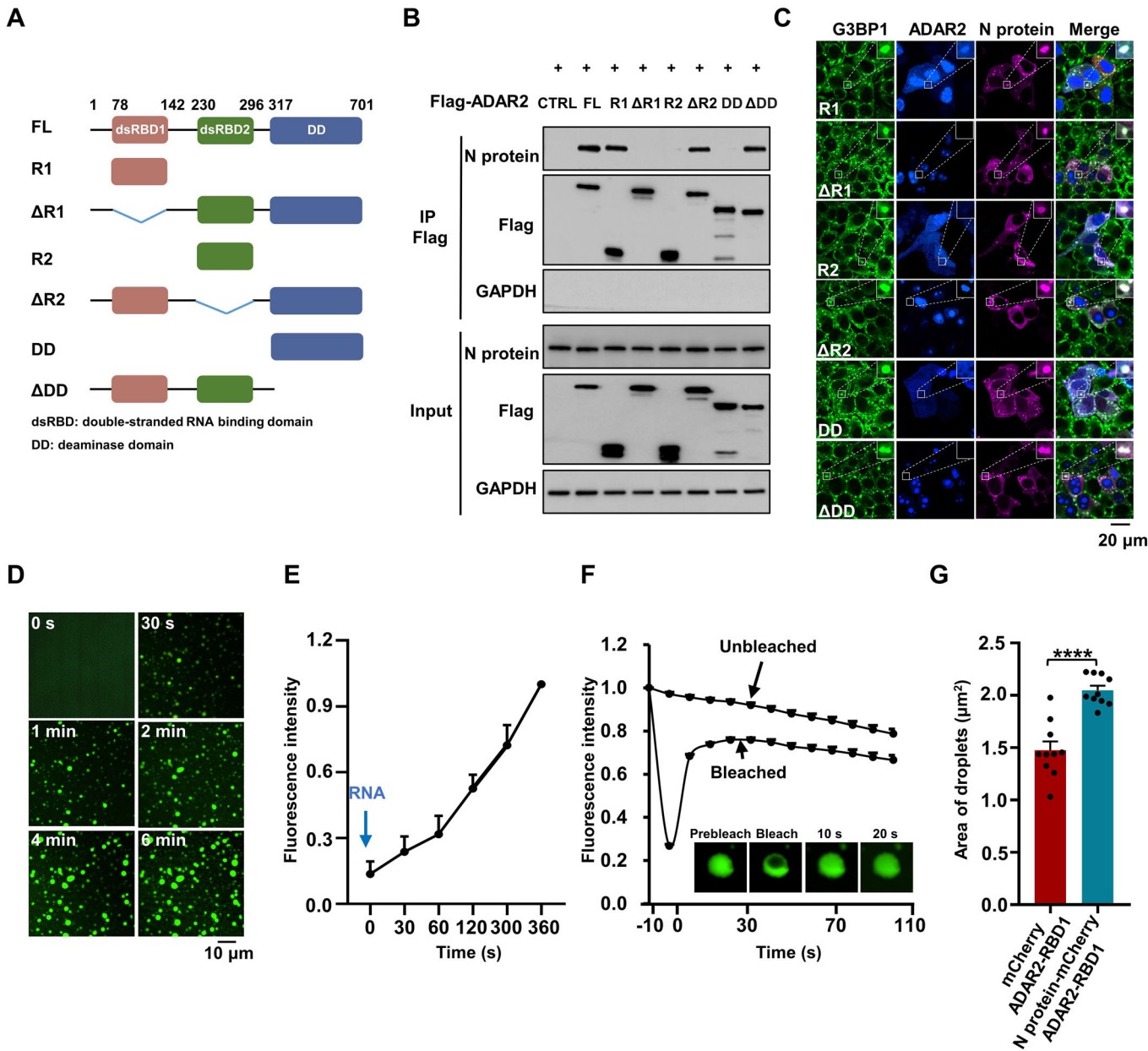

**Figure 5. The RNA binding domain is essential for ADAR2 association with the N protein to form phase separation-mediated condensates.**

(A) Schematic domain structure of ADAR2. (B) Characterization of ADAR2 mutants required for the interaction with the N protein. HeLa cells were co-transfected with plasmids encoding the N protein and either Flag control or various Flag-tagged ADAR2 truncations. (C) Characterization of ADAR2 mutants required for SG localization. HeLa cells co-transfected with the N gene and BFP-tagged ADAR2 mutants were treated with AS for 45 min to induce SG formation, followed by immunostaining for the N protein and G3BP1. (D) Time-lapse microscopy of RNA-induced RBD1 phase separation. Liquid droplets formed upon mixing of ADAR2-RBD1 protein with 75 ng/μL RNA. (E) Quantification of fluorescence intensity of ADAR2-RBD1 liquid droplets in the presence of 75 ng/μL RNA. Values and error bars were represented as the mean ± SEM of five independent biological replicates. (F) FRAP analysis of ADAR2-RBD1 liquid droplets formed in the presence of RNA. The dotted square displays the photobleached region. Values represent mean ± SEM from $n = 30$ droplets. Scale bar, 10 μm. (G) Column scatter charts displaying the droplet area from experiments related to Appendix Fig. S11B. Values and error bars were represented as the mean ± SEM of 10 independent images. Statistical analysis was performed with a two-tailed unpaired t-test. ****$P < 0.0001$. Source data are available online for this figure.

RNA sequence analysis from COVID-19 patients (Di Giorgio et al, 2020; Wang et al, 2020a). Host deaminases likely engage in RNA editing of viral RNA at various stages throughout the viral infection process, including 'early' editing of viral genomes and negative-sense transcripts before viral replication and 'late' editing of positive/negative-strand transcripts after viral replication. Furthermore, the deaminases overexpression indeed increased SARS-CoV-2 RNA mutagenesis and impaired viral production. These results demonstrated that deaminases function as a host antiviral response, inducing extensive mutations in SARS-CoV-2 RNA during the

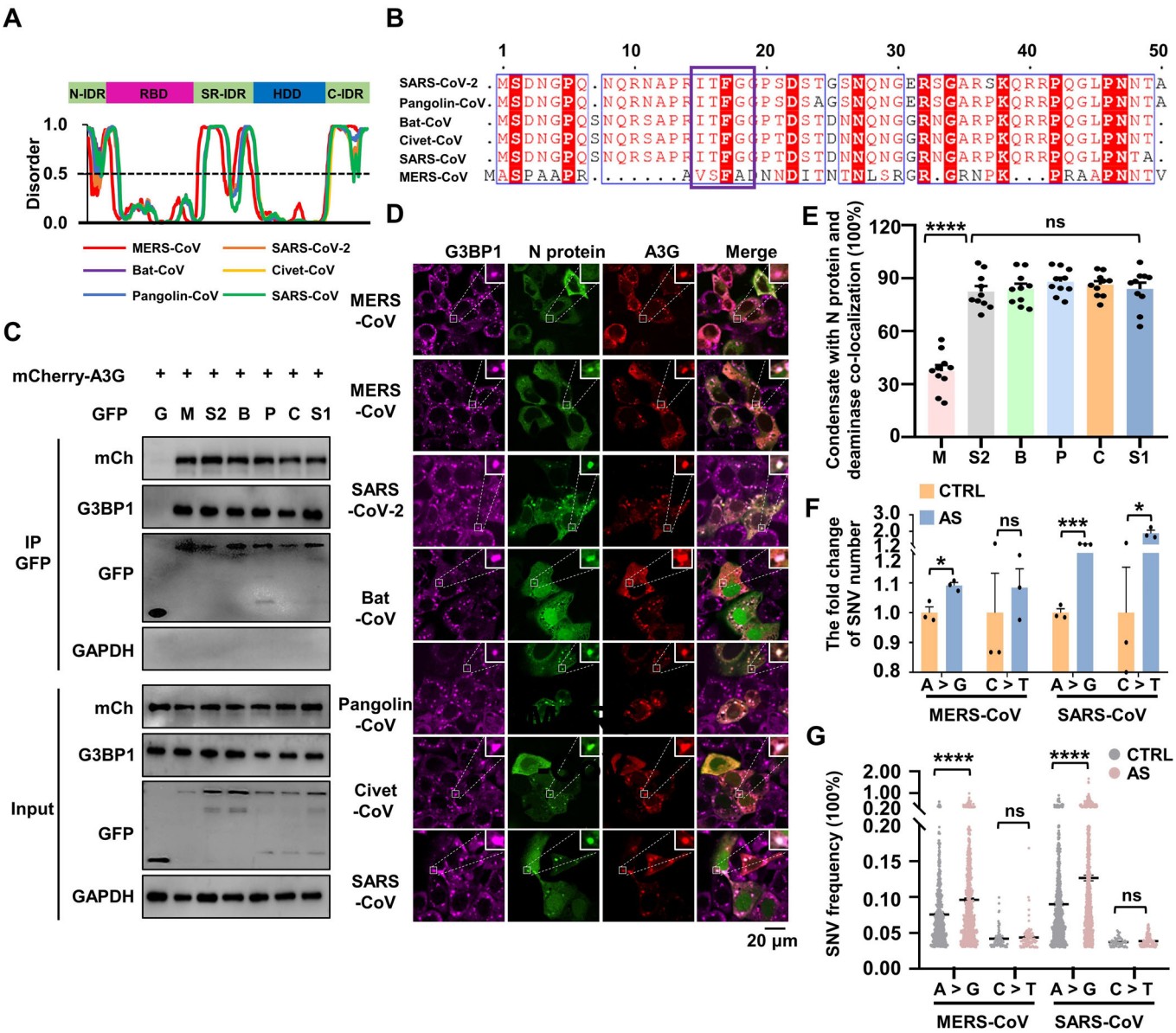

Figure 6. Functional characteristics of coronavirus N proteins.

(A) Domain structure (top) and sequence features (bottom) of the N proteins from MERS-CoV, SARS-CoV-2, bat-CoV, civet-CoV, pangolin-CoV and SARS-CoV. The domains of the N proteins were aligned with the disorder propensity calculated by the DISOPRED3 server. (B) Sequence alignment of six related coronavirus N proteins (MERS-CoV, SARS-CoV-2, bat-CoV, civet-CoV, pangolin-CoV and SARS-CoV) showing the N terminal IDR. (C) Interaction assay between the N proteins and APOBEC3G (A3G). HeLa cells were co-transfected with plasmids encoding GFP-tagged N proteins from MERS-CoV, SARS-CoV-2, bat-CoV, civet-CoV, pangolin-CoV and SARS-CoV or GFP control, and mCherry-tagged A3G. (D) N proteins from MERS-CoV, SARS-CoV-2, bat-CoV, civet-CoV, pangolin-CoV and SARS-CoV exhibit the ability to co-localize with A3G in SGs in HeLa cells. Cells co-transfected with the N protein and A3G were treated with AS for 45 min to induce SG formation, followed by immunostaining for N protein, A3G and endogenous G3BP1. Scale bar: 20 μm. (E) Quantification of condensates with co-localization of N protein and deaminase in fixed HeLa cells. Values and error bars were represented as the mean ± SEM of ten independent biological replicates. Statistical analysis was performed with a one-way ANOVA test. ns > 0.05, ****$P$ < 0.0001. ns, not significant. (F) Fold change of SNVs number in the N protein transcriptome (MERS-CoV and SARS-CoV) in cells treated with or without AS. Values and error bars were represented as the mean ± SEM of three independent biological replicates (allelic fraction ≥0.03%). Statistical analysis was performed with an unpaired t-test. ns > 0.05, *$P$ < 0.05, ***$P$ < 0.001, ns, not significant. The number of SNVs with A > G/C > T mutations was normalized to the number of control group. (G) SNVs frequency in the N protein transcriptome (MERS-CoV and SARS-CoV) in cells treated with or without AS. Values and error bars were represented as the mean ± SEM of three independent biological replicates. Statistical analysis was performed with an unpaired t-test. ns >0.05, ****$P$ < 0.0001. ns not significant. Source data are available online for this figure.

early stages of infection, potentially leading to viral RNA degradation and inhibition of viral RNA replication. Conversely, the number of SNVs induced by APOBECs or ADARs on the SARS-CoV-2 genome was significantly decreased after several passages, suggesting that deaminases activity may be restrained during the late phase of SARS-CoV-2 infection. One possible mechanism is that SARS-CoV-2 encodes a defense protein to counteract the host antiviral response. A similar process has been reported for APOBEC3G, a potent arm of the host defense against HIV and other retroviruses by introducing "typographical errors" during viral replication (Esnault et al, 2005; Mangeat et al, 2003; Sheehy et al, 2002; Sheehy et al, 2003; Yu et al, 2003). In turn, HIV encodes the virion infectivity factor (Vif), which induces A3G degradation and protects the virus from A3G-induced inactivation (Sheehy et al, 2002; Sheehy et al, 2003; Yu et al, 2003). However, the SARS-CoV-2-encoded protein to counteract host deaminases remains unknown and requires further investigation.

Given that some viral infections can trigger the Type I interferon (IFN-I) response, thereby enhancing the expression of ADAR1p150, ADAR1 is widely regarded as a key instigator of RNA virus mutagenesis (Borden & Williams, 2011; McNab et al, 2015). However, the precise extent to which this upregulation serves as the sole modulator of A to I transition in viral RNA has not been fully elucidated. Indeed, ADAR2 also contributes to host immune responses and influences RNA virus infection through its deaminase activity, as demonstrated in cases such as BoDV, HIV-1 and HDV5 (Doria et al, 2011; Jayan & Casey, 2002; Tomaselli et al, 2015; Yanai et al, 2020). In our study, we established that both ADAR1 and ADAR2 are involved in the RNA editing process of the SARS-CoV-2 genome and significantly impact SARS-CoV-2 production. Thus, ADAR1 and ADAR2 share similar characteristics and act as host antiviral factors to regulate viral replication and mutagenesis.

An intriguing observation is that the G > U (G > T) and C > A transversions represent the second most prevalent groups of mutations for nucleotides C and G, potentially due to mutagenic activity of ROS produced by cells infected with SARS-CoV-2 (Laforge et al, 2020; Mourier et al, 2021; Shang et al, 2022; Smith, 2017). Additionally, the A > U (A > T) and U > A (T > A) substitutions on SARS-CoV-2 genomes appear as the second most common groups for nucleotides A and T change. Similarly, A > G and C > T transitions are the predominant types of mutations in the mRNA of the N protein. G > A and T > C changes, along with the A > T transversion, exhibit the second most dominant mutational signatures in the transcripts of the N protein. This suggests that additional factors may regulate the mRNA editing process. Such non-classic alterations have also been observed and characterized in mammalian transcripts (Grohmann et al, 2010; Klimek-Tomczak et al, 2006; Li et al, 2012; Li et al, 2011; Niavarani et al, 2015; Sharma et al, 1994; Tao et al, 2021). Furthermore, A3A has been shown to regulate G > A mRNA editing in Wilms Tumor 1 (Niavarani et al, 2015). Transamination and transglycosylation mechanisms have been proposed to underlie U to C editing events in plant transcripts (Castandet and Araya, 2011; Gerke et al, 2020; Knoop, 2023). However, the mechanistic basis for this non-classic RNA editing remains largely unknown and warrants further investigation. Consequently, the mutational landscape of the SARS-CoV-2 genome is shaped by multi-driving forces from both the host and the virus.

Previous studies have indicated that ADAR2 is predominantly localized to the nucleoplasm and nucleolus, even under cellular stress conditions (Hajji et al, 2022; Weissbach and Scadden, 2012), which appears to contrast with our findings. This inconsistency might arise from the low abundance of cytoplasmic ADAR2, which could be below the threshold of detection, thus obscuring its presence in the cytoplasm and SGs during stress. Emerging evidence, however, has shown that while ADAR2 is primarily nucleoplasmic and nucleolar, a minor fraction is also expressed in the cytoplasm (Aizawa et al, 2010; Behm et al, 2017; Jimeno et al, 2021; Marcucci et al, 2011). To further confirm the subcellular distribution of ADAR2, we employed a variety of experimental approaches to verify its presence in the cytoplasm. Firstly, we performed an immunofluorescence assay to detect both exogenous and endogenous ADAR2 in the cytoplasm (Figs. 2B, 3A, 4E, 5C, EV2A–C, EV3D and EV4C; Appendix Fig. S7E, S10C and S12A). Next, subcellular fractionation coupled with immunoblotting assays verified the presence of both exogenous and endogenous ADAR2 in the cytoplasm, although it remains mainly in the nucleoplasm and nucleolus (Appendix Fig. 6A,B). Finally, we knocked down ADAR2 expression in HeLa cells and conducted immunofluorescence staining with an anti-ADAR2 antibody to assess the specificity of the signal. Quantitative analysis of immunofluorescence signals indicated a significant reduction of approximately 85% in fluorescence intensity following shRNA-mediated knockdown of ADAR2 (Appendix Fig. 6C,D). This reduction confirms the antibody's specificity for ADAR2 and further supports the presence of ADAR2 in cytoplasm.

In the context of RNA-protein interactions, the SARS-CoV-2 N protein, responsible for binding the viral RNA, has been identified as the most significantly enriched viral protein in SARS-CoV-2 RNA purifications (Schmidt et al, 2021). Additionally, ADARs have been identified as binders of SARS-CoV-2 RNA among host cell proteins (Schmidt et al, 2021). In terms of protein-protein interactions, A3G has been reported to associate with the coronavirus N protein, facilitating its packaging into SARS-CoV virus-like particles (Wang and Wang, 2009). Furthermore, a SARS-CoV-2 protein interaction map has revealed that the N protein interacts with several RNA-binding host proteins (Gordon et al, 2020). Despite these indications that deaminases may target viral RNA, the underlying mechanism has remained unclear. In this study, we demonstrate that the N protein, unlike other SARS-CoV-2 viral RNA-binders, specifically interacts with a variety of host deaminases, including A1, AID, A3D, A3F, A3G, A3H, ADAR1 and ADAR2. These findings suggest that the N protein serves as a specific bridging molecule for host deaminases and viral RNA, linking them together. Within the APOBEC family, only A1, A3A and A3G have previously been shown to possess RNA editing activities (Sharma et al, 2016; Sharma et al, 2015; Teng et al, 1993; Wolfe et al, 2019). Consistent with these reports, A1, A3G, ADAR1 and ADAR2 indeed targeted viral RNA and induced deamination on the SARS-CoV-2 genome. In addition, AID, A3D and A3H exhibited the potential to bind RNA and edit the viral RNA. Further investigation is warranted to elucidate the RNA binding specificity of these host deaminases.

Membraneless organelles offer a strategic means of concentrating specific proteins and nucleic acids, thereby enhancing local concentrations and significantly strengthening binding affinities (Alberti, 2017; Antoniou and Schwartz, 2016; Kent et al, 2020;

Tauber et al, 2020). In line with this, we observed that the formation of N protein-deaminases condensates facilitates more robust ADAR2 binding to N protein RNA. The biological role of these molecularly enriched condensates is to expedite the reaction rate (Antoniou and Schwartz, 2016). Notably, we demonstrated that the N protein infiltrates deaminase-rich RNA granules, forming N protein-deaminases condensates that substantially enhance the RNA mutational rate of the N protein. In contrast, the $N^{F17A}$ mutant failed to be recruited into deaminase-rich RNA granules, resulting in a marked reduction in deaminase RNA editing efficiency. The underlying mechanism appears to involve RNA granules, which, as condensed structures, elevate the local concentration of viral RNA and deaminases, thereby boosting deamination activity. These findings suggest that phase separation may serve as a regulatory switch to govern deamination activity, presenting a novel organizational strategy to enhance viral RNA editing efficiency. However, phase separation-mediated condensation often acts as a double-edged sword in regulating biomolecule activity. Phase separation can create a selective environment that restricts the catalytic function of the cytosolic exonuclease TREX1 (Zhou et al, 2021). Moreover, phase separation-induced compact chromatin can facilitate the recruitment of repressive factors, leading to gene expression silencing (Guo et al, 2021). RNA polymerization can be inhibited in the polyamine-containing condensation (Drobot et al, 2018). Conversely, polyethylene glycol (PEG)/dextran aqueous two-phase system (ATPS)-mediated phase separation greatly increases RNA catalytic reactions (Strulson et al, 2012). Phase-separation protein domains can also serve as enhancers to boost CRISPRa activity (Liu et al, 2023). Therefore, elucidating the physicochemical properties of micro-condensates, including size, chemical composition, and structural stability, is essential for predicting deamination activity.

In summary, our study presents experimental evidence that host deaminases serve as antiviral host factors, inducing mutations within the SARS-CoV-2 genome. The N protein plays a pivotal role in this process, acting as a molecular bridge to direct deaminases towards viral RNA. Additionally, the virus appears to hijack host RNA granules to promote deaminases-mediated RNA editing activity, potentially driving its own evolution. Our findings may aid in forecasting viral mutation patterns and offer novel perspectives for the development of therapeutic strategies against SARS-CoV-2.

# Methods

## Plasmid construction

DNA fragments encoding the APOBEC family proteins, ADAR family proteins and SARS-CoV-2 proteins, as described in this manuscript, were synthesized by GenScript and subsequently amplified via PCR using Phanta® Max Super-Fidelity DNA Polymerase (Vazyme, P505-d1). The coding regions of ADAR2 deletion mutants were generated by PCR amplification from a plasmid harboring the full-length ADAR2 sequence, employing a suitable set of primers. Exnase (Vazyme, C214-02-AF) was subsequently utilized to clone these sequences into one of the following vectors: CMV-mCherry-Flag, CMV-mCherry-HA, CMV-BFP-Flag, CMV-BFP-HA, CMV-GFP, CMV-Flag. All plasmid inserts were confirmed by BioSune Sanger sequencing.

## Cell culture, transfection, and harvest

HeLa cells were cultured in Dulbecco's Modified Eagle Medium (DMEM) supplemented with 10% bovine growth serum (FBS) and 1% penicillin/streptomycin at 37 °C in a humidified incubator with 5% $CO_2$. For plasmid transfection, cells were plated in 6-well plates and transfected at approximately 70% confluence using the Lipofectamine 3000 reagent (Invitrogen, L3000008). At 48 h post-transfection, cells were harvested based on fluorescence activating cell sorter (FACS) criteria, as dictated by the fluorescence emitted by the transfected plasmid.

## Generation of G3BP1/2 knockout cell lines

HeLa cells were seeded in 24-well plates and transfected with 300 ng sgRNA and 700 ng spCas9 plasmid using the Lipofectamine 3000 transfection kit (Invitrogen, L3000015). At 48 h post-transfection, cells were sorted into 96-well plates using fluorescence-activated cell sorting (FACS). Genomic DNA from the edited cells was extracted using QuickExtract™ DNA Extraction Solution (Lucigen, QE09050), and the target sequences were amplified by PCR and subjected to BioSune Sanger sequencing for analysis. Individual clones were lysed in RIPA buffer (Beyotime, P0013B) containing 1 mmol/L phenylmethanesulfonyl fluoride (PMSF) and 1% (v/v) protease inhibitor cocktail (Thermo Fisher Scientific, 78443) on ice for 15 min. The lysate was centrifuged at 12,000 rpm for 15 min at 4 °C. The supernatant was harvested for SDS-PAGE analysis using G3BP1 antibody (Proteintech, 66486-1-Ig) and G3BP2 antibody (Proteintech, 16276-1-AP). A list of sgRNAs utilized in this study is provided in Appendix Table S1.

## Generation of ADAR2 knock down cell lines

A specific short hairpin RNA (shRNA) construct was designed to target human ADAR2 and cloned into the pLKO.1 lenti-vector. HeLa cells with reduced ADAR2 expression were generated using shRNA lentiviral particles and selected with puromycin selection. The shRNA sequences utilized in this study was 5′- cggagatccttgctcagattt-3′.

## Co-immunoprecipitation

HeLa cells were transfected with the indicated plasmids using the Lipofectamin 3000 reagent (Invitrogen, L3000008). At 48 h post-transfection, cells were collected and lysed in RIPA buffer (Beyotime, P0013B) supplemented with 1 mmol/L PMSF and 1% (v/v) protease inhibitor cocktail (Thermo Fisher scientific, 78443). Lysates were incubated on ice for 15 min and centrifuged at 16,000 × $g$ at 4 °C for 20 min. The supernatant was then incubated with pre-equilibrated anti-Flag beads (Bimake, B26101) or anti-GFP beads (KT Health, KTSM1333) for 4 h at 4 °C. Beads were washed three times with PBS containing 0.05% (V/V) Tween-20 to remove nonspecific bindings. Subsequently, 1× loading buffer with SDS was added to each sample and heated at 98 °C for 10 min to elute bound proteins. The supernatant was collected and subjected for SDS-PAGE analysis.

## RNA immunoprecipitation (RIP) assays

HeLa cells were transfected with plasmids encoding the N WT or $N^{F17A}$ mutant DNA fragments. RIP assays were conducted using

RIP assay kit (Geneseed Biotech, P0102) following the manufacturer's protocol with the following antibodies: rabbit anti-ADARB1 (Proteintech, 22248-1-AP) and normal rabbit IgG (Abcam, ab190475). The immunoprecipitated RNA was isolated and analyzed by reverse transcription quantitative PCR (RT-qPCR), with primer sequences detailed in Appendix Table S2.

## Synthesis of RNA by in vitro transcription (IVT)

For the production of IVT mRNA, template plasmids containing the wide-type N gene or the F17A variant from SARS-CoV-2, MERS-CoV and SARS-CoV were subjected to linearization, followed by in vitro transcription using T7 RNA polymerase from the T7 High Yield RNA transcription kit (Novoprotein, E131). The resulting linear mRNA was capped and purified by the Capping System (Novoprotein, M082). The integrity of all mRNA was confirmed by agarose gel and electrophoresis prior to cryopreservation at −20 °C.

## IVT mRNA transfection

HeLa cells were plated in 12-well plates and transfected with IVT mRNA at approximately 70% confluence using the Lipofectamine MessengerMax (Invitrogen, LMRNA001) following the manufacturer's protocol.

## Immunofluorescence

HeLa cells were seeded to achieve 70% confluence and transfected with plasmids in 24-well glass bottom plates (Cellvis, Mountain View, USA). Oxidative stress was induced with 200 µmol/L sodium arsenite (Sigma Aldrich, S7400) for 45 min. ER stress was triggered by treatment with 2 mmol/L DTT for 1 h. Viral stress mimicry was performed by transfecting 2 µg polyI:C using the Lipofectamine 3000 (Invitrogen, L3000015) for 7 h. Osmotic stress was induced using 0.5 mol/L sorbitol (Sigma Aldrich, S1876) for 1 h. Cells were fixed with 4% paraformaldehyde for 15 min at room temperature and rinsed three times with PBS. After a 2-hour incubation with blocking buffer (5% normal goat serum (Bioss Antibodies, C-0005), 0.3% Triton X-100 in PBS) at room temperature, cells were incubated with primary antibodies against ADAR1 antibody (Proteintech, 14330-1-AP; 1:200 dilution) and ADARB1 antibody (Proteintech, 22248-1-AP; 1:200 dilution), G3BP1 (Proteintech, 66486-1-Ig; 1:200 dilution), TIA1 (Proteintech, 12133-2-AP; 1:200 dilution) or Flag (Cell Signaling Technology, 14793 s; 1:1000 dilution) overnight at 4 °C. Following three washes in PBS, cells were incubated with Alexa Fluor 488/647 conjugated secondary antibodies for 1 h. Nuclei were counterstained with 4,6-diamidino-2-phenylindole (DAPI; Yeasen, 40728ES03). Images were captured with a Nikon spinning disk confocal microscope.

## Preparation of cytoplasmic and nuclear extracts

Cytoplasmic and nuclear fractionation was conducted using the Nuclear and Cytoplasmic Protein Extraction Kit (Beyotime Biotechnology, P0028) following the manufacturer's instructions. Protein concentrations in cytoplasmic and nuclear fractionations were determined using the BCA Protein Assay Kits (Thermo Scientific, 23225). GAPDH (Absin, abs132004; 1:5000 dilution) served as a cytoplasmic protein marker, and Lamin A/C (Proteintech, 10298-1-AP, 1:5000 dilution) as a nuclear protein marker.

## Western blotting

Equal amounts of protein samples were resolved by SDS-PAGE and transferred onto methanol-activated polyvinylidene fluoride (PVDF) membranes (Millipore, IPVH00010). Membranes were blocked for 1 h at room temperature in Tris-buffered saline and 0.1% Tween-20 (TBST) containing 5% (w/v) nonfat milk. Subsequently, the membranes were incubated with primary antibodies at dilution ratio recommended by the manufacturers at 4 °C. After three washes in TBST, protein bands were detected with horse radish peroxidase (HRP)-conjugated secondary antibodies and Immobilon Western enhanced chemiluminescent solution (Millipore, WBKLS0100). Protein levels were normalized by probing the blots with GAPDH antibody (Absin, abs132004; 1:5000 dilution).

## SARS-CoV-2 GFP/ΔN trVLP infection

Caco-2 cells expressing WT N protein were infected with SARS-CoV-2 GFP/ΔN P1 trVLP (Ju et al, 2021) at a multiplicity of infection (MOI) of 0.1. After a 24-hour infection, cells are rinsed with Dulbecco's Phosphate-Buffered Saline (DPBS) and lysed with Trizol reagent (Invitrogen, 15596018). Uninfected Caco-2 cells expressing WT N protein served as mock controls.

## SARS-CoV-2 GFP/ΔN trVLP passage

SARS-CoV-2 GFP/ΔN P0 trVLPs (Ju et al, 2021) are used to infect Caco-2 cells expressing either WT N protein or F17A mutant N protein at an MOI of 0.1 for 24 h. The supernatants (P1) were collected and used to infect Caco-2 cells expressing WT N protein or F17A mutant N protein. Cell cultures from each successive passage on Caco-2-N WT or F17A mutant cells were designated as P1 to P6, respectively. Cells are rinsed with DPBS and lysed with Trizol reagent (Invitrogen, 15596018).

## SARS-CoV-2 mutation analysis

Viral RNA from the supernatant was extracted using Trizol reagent (Invitrogen, 15596018). Complementary DNA (cDNA) was synthesized from the RNA using the HiScript III 1st Strand cDNA Synthesis Kit (Vazyme, R312-01). Sequencing libraries were prepared using the TruePrepTM DNA Library Prep Kit V2 for Illumina (Vazyme, TD503-01) and sequenced by Illumina Hiseq X Ten platform. The RNA-seq data were aligned to the reference genome of SARS-CoV-2 (NC_045512), SARS-CoV (NC_004718) and MERS-CoV (NC_019843) using the STAR software (Version 2.5.1).

Plasmids containing SARS-CoV-2 gene fragments were transfected into HeLa cells. At 48 h post-transfection, cells were treated with 120 µmol/L AS to induce SG formation for 10 h. Cells were then harvested, and total RNA was extracted using Trizol reagent (Invitrogen, 15596018). 1 µg of RNA was reversely transcribed into cDNA using Reverse Transcriptase (Vazyme, R223-01-AB). Target regions were amplified with Phanta Max Super-Fidelity DNA Polymerase (Vazyme, P505-d1) and subjected to high-throughput sequencing on the Illumina Hiseq X Ten platform. Preprocessed

data were aligned to the plasmid sequences using bwa software (Version 0.7.17).

All variants from RNA-seq and deep-seq were identified and quantified using bam-readcount with parameters −q 20 −b 30. SNVs were filtered based on a minimum threshold of 20 reads.

## Protein purification

DNA fragments encoding proteins of interest were cloned into the pET-28(a) vector. Proteins were expressed in E. coli BL21 (DE3) (Trans, CD601) and induced with 1.0 mmol/L IPTG at 16 °C for 14 h. Cell lysates were collected by centrifugation and then resuspended in buffer A (20 mM Tris-HCl, pH 8.0, 500 mM NaCl, 10% (v/v) glycerol, 1 mM PMSF). The supernatant was treated with 0.4 μg/mL DNase and RNase to remove any contaminating DNA and RNA. The protein of interest was purified by Ni resin (TaKaRa, 635660) and eluted by buffer B (20 mM Tris-HCl, pH 8.0, 500 mM NaCl, 10% (v/v) glycerol, 500 mM imidazole). Further purification was performed on an AEKTA purifier (GE Life Sciences). Purified proteins were analyzed by SDS-PAGE, and stored in buffer C (20 mM Tris (pH 8.4), 500 mM NaCl, 10% (v/v) glycerol) at −80 °C before subsequent analysis. Recombinant mCherry fusion proteins (mCherry-N protein FL and mCherry-N$^{F17A}$ protein) and mEGFP fusion proteins (mEGFP-G3BP1 and mEGFP-ADAR2-RBD1) were purified following the same protocol.

## In vitro droplet assay

For assessing RNA-dependent droplet formation, total RNA extracted from HeLa cells using Trizol reagent (TaKaRa, 9108) was co-incubated with purified SARS-CoV-2 N protein, G3BP1 or ADAR2-RBD1. To evaluate the impact of the F17A mutation in the N protein on the phase separation of N protein/G3BP in vitro, mCherry-tagged full-length N protein (mCherry-N protein FL) or mCherry-tagged N protein F17A mutant (mCherry-N$^{F17A}$ protein) was co-incubated with mEGFP-G3BP1 in the presence of HeLa cellular RNA. The formation of LLPS droplets was observed under a Nikon Spinning Disk microscope equipped with a 60× oil immersion objective using a 96-well glass bottom plate (Cellvis, P96-1.5H-N).

## Fluorescence recovery after photobleaching (FRAP) analysis

FRAP analysis was performed using a Nikon Spinning Disk microscope equipped with two laser systems. 1–2 images were captured before the center of the indicated protein droplets was bleached with a 488 nm laser. Following photobleaching, images were acquired at 10-s intervals for 2 min.

## Statistical analysis

All statistical analyses were performed using GraphPad Prism 8 (GraphPad Software, San Diego, USA) or Microsoft Excel (Professional 2019, Microsoft Corporation, Redmond, USA). The results of all statistical tests, including the number of samples and $P$ values, were shown in the corresponding figure legends. Results were presented as mean ± SEM. The significance of $P$ values is represented as follows: ns > 0.05, $*P < 0.05$; $**P < 0.01$, $***P < 0.001$, $****P < 0.0001$, ns, not significant.

## Data availability

The sequence data for the RNAseq experiment reported in this study has been deposited into the BioProject, with accession number PRJNA824251 (https://www.ncbi.nlm.nih.gov/bioproject/PRJNA824251).

The source data of this paper are collected in the following database record: biostudies:S-SCDT-10_1038-S44318-024-00314-y.

## Peer review information

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

## Acknowledgements

We thank all members of Huang lab for helpful discussions and Dr. Haitao Yang from ShanghaiTech University for insightful discussion; Molecular Imaging Core Facility, Molecular and Cellular Biology Core Facility and Discovery Technology Platform of SIAIS for support with imaging and cellular biology experiments. This work is supported by the National Natural Science Foundation of China (82302497), the National Key R&D Program of China (2021YFA0804702), the National Natural Science Foundation of China (82188102), the Foundation of Science and Technology Commission of Shanghai Municipality (22DX1900400), the leading talents of Guangdong province program (2016LJ06S386), the National Natural Science Foundation of China (22177073), the National Natural Science Foundation of China (32070153), the Tsinghua University Spring Breeze Fund (2021Z99CFY030) and the Beijing Municipal Natural Science Foundation (M21001).

## Author contributions

**Zhean Li**: Resources; Formal analysis; Methodology; Writing—review and editing. **Lingling Luo**: Resources; Formal analysis; Methodology; Writing—review and editing. **Xiaohui Ju**: Resources; Formal analysis; Methodology; Writing—review and editing. **Shisheng Huang**: Resources; Software; Formal analysis; Methodology. **Liqun Lei**: Resources; Methodology. **Yanying Yu**: Resources; Methodology. **Jia Liu**: Resources. **Pumin Zhang**: Resources. **Tian Chi**: Resources. **Peixiang Ma**: Resources; Funding acquisition; Methodology. **Cheng Huang**: Conceptualization; Resources; Supervision; Writing—review and editing. **Xingxu Huang**: Conceptualization; Resources; Supervision; Funding acquisition; Writing—review and editing. **Qiang Ding**: Conceptualization; Resources; Supervision; Funding acquisition; Writing—review and editing. **Yu Zhang**: Conceptualization; Supervision; Funding acquisition; Validation; Writing—original draft; Project administration.

Source data underlying figure panels in this paper may have individual authorship assigned. Where available, figure panel/source data authorship is listed in the following database record: biostudies:S-SCDT-10_1038-S44318-024-00314-y.

## Disclosure and competing interests statement

The authors declare no competing interests.

# Expanded View Figures

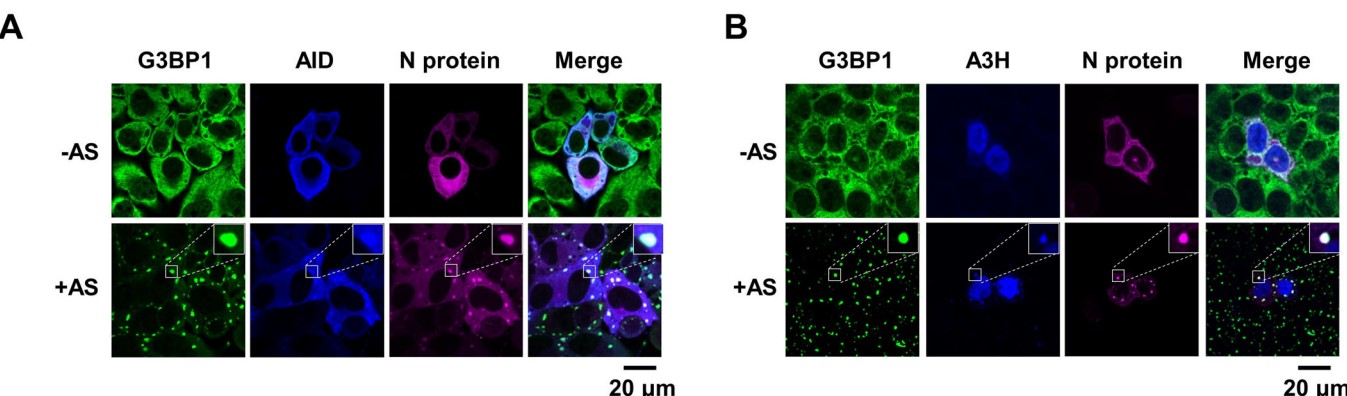

**Figure EV1.   Co-localization of N Protein with AID and A3H in SGs.**

(**A**) AID co-localizes with N protein in SGs in HeLa cells. HeLa cells were transfected with the N gene and AID, then treated with AS for 45 min to induce SG formation, followed by immunostaining for N protein, AID and G3BP1. Scale bar: 20 μm. (**B**) A3H co-localizes with N protein in SGs in HeLa cells. HeLa cells were transfected with the N gene and A3H, then treated with AS for 45 min to induce SG formation, followed by immunostaining for N protein, A3H and G3BP1. Scale bar: 20 μm.

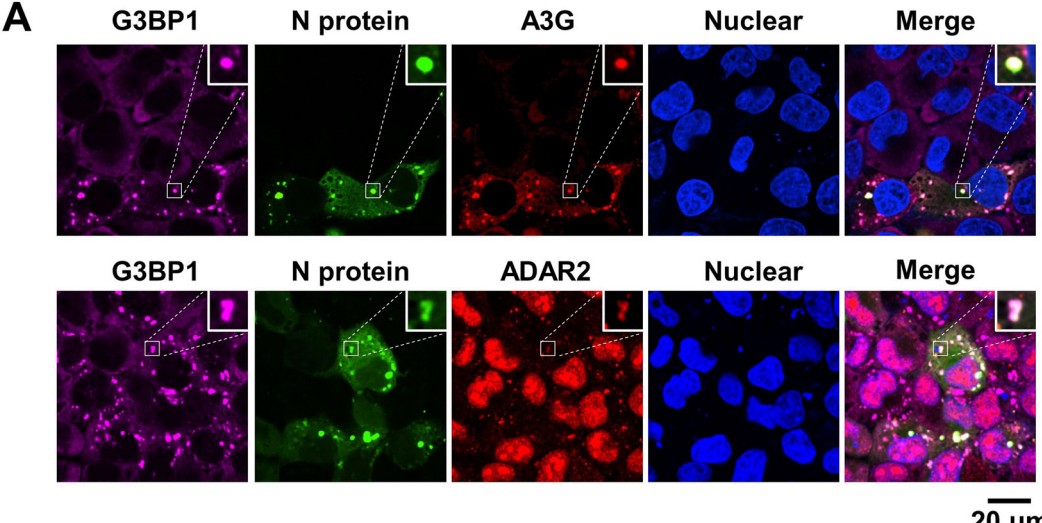

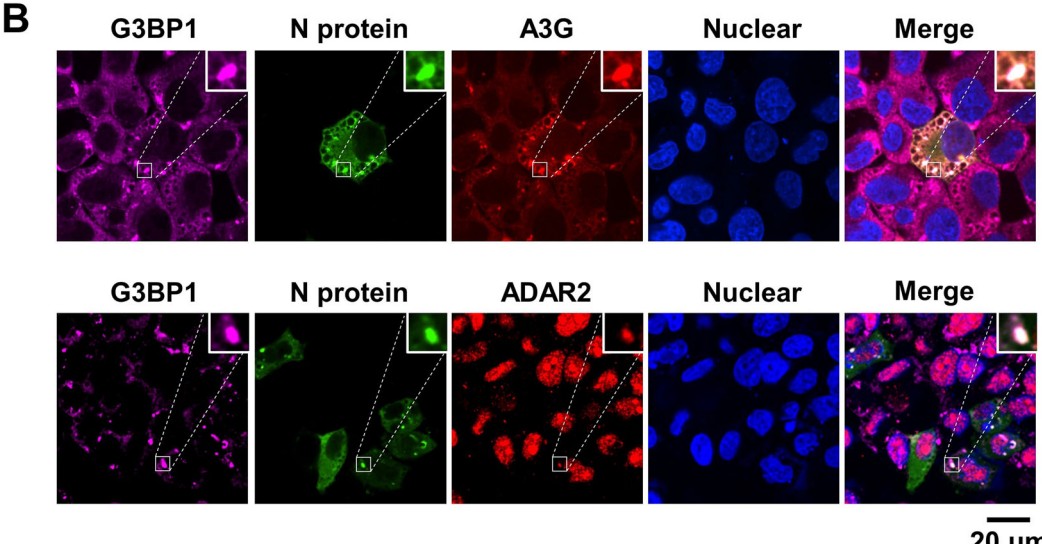

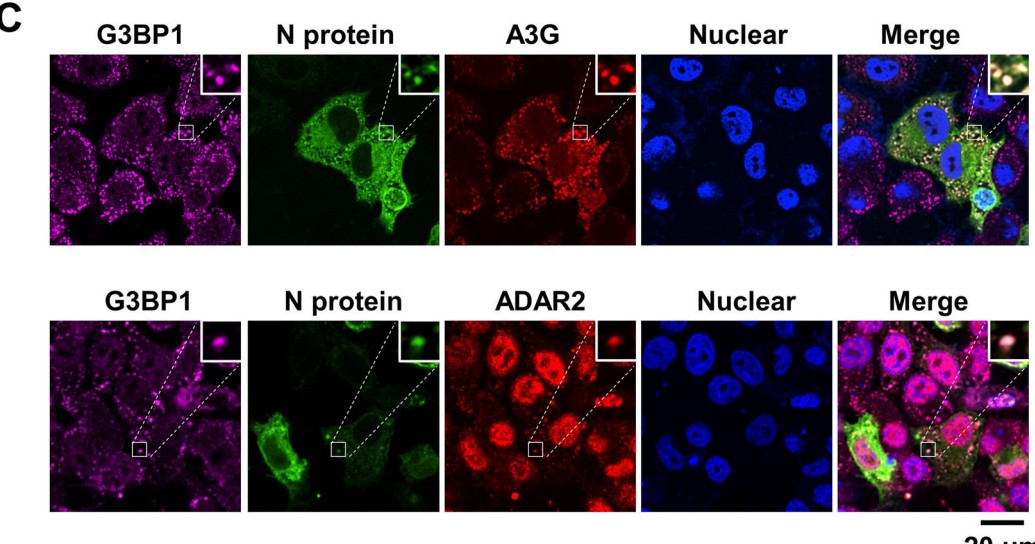

◀  **Figure EV2.  Specific co-localization of N protein with host deaminases under diverse stress conditions.**

(**A–C**) A3G or ADAR2 co-localizes with N protein in SGs in response to various stressors. HeLa cells were transfected with the N gene and A3G or ADAR2, and treated with polyI:C for 7 h (**A**), DTT for 1 h (**B**) or sorbitol for 1 h (**C**) to induce SG formation, followed by immunostaining for N protein, A3G or ADAR2 and G3BP1. Scale bar: 20 μm.

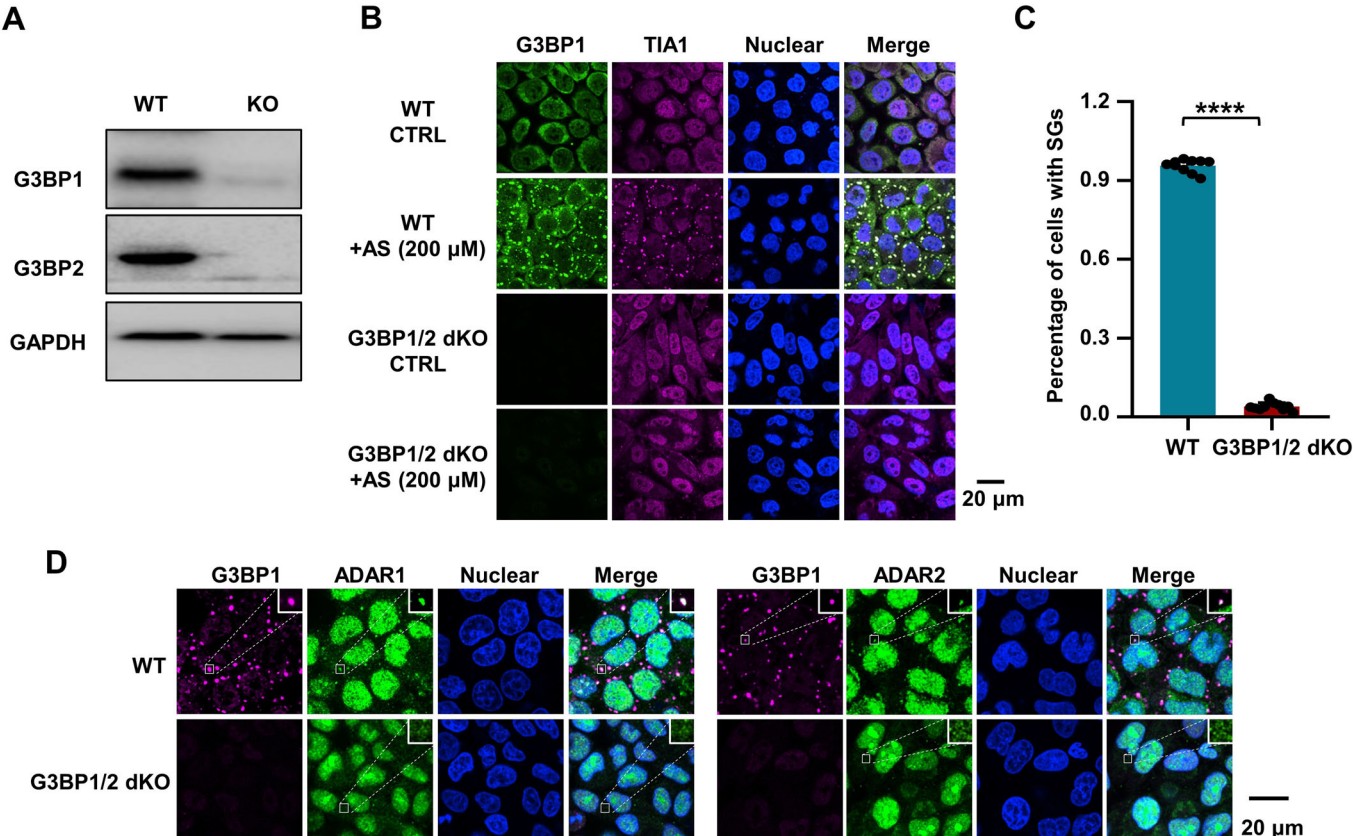

**Figure EV3.   The lack of G3BP1/2 disrupts the formation of N protein-deaminase complex-containing RNA condensates.**

(**A**) CRISPR-Cas9-mediated G3BP1/2 knockout in HeLa cells. (**B**) G3BP1/2 dKO HeLa cells fail to form SGs under AS-induced stress. HeLa cells with or without G3BP1/2 dKO were treated with AS for 45 min to induce SG formation, followed by immunostaining for the endogenous G3BP1 (GFP) and endogenous TIA1 (red). Scale bar: 20 μm. (**C**) Quantification of SGs, expressed as the percentage of cells containing SGs in fixed HeLa cells with or without G3BP1/2 dKO. Data are shown as means ± SEM ($n = 10$ independent images). Statistical analysis was performed with a two-tailed unpaired t-test. ****$P < 0.0001$. (**D**) Depletion of G3BP1/2 abolishes the localization of ADAR1/2 in SGs under stress. HeLa cells with or without G3BP1/2 dKO were treated with AS for 45 min to induce SG formation, followed by immunostaining for endogenous G3BP1 and endogenous ADAR1 or ADAR2. Scale bar: 20 μm. Source data are available online for this figure.

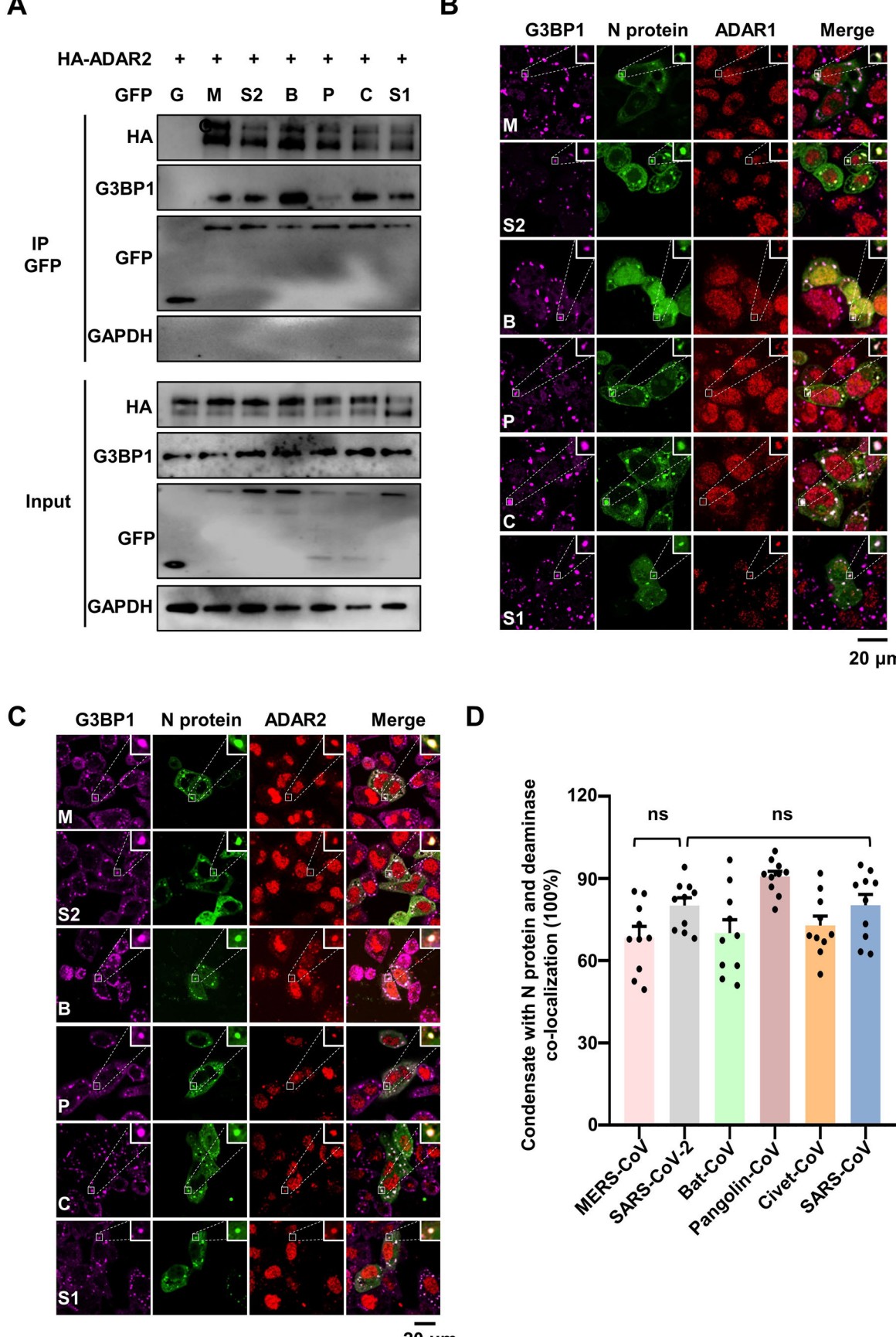

◀

**Figure EV4. Functional characteristics of coronavirus N protein.**

(A) Interaction assay between N protein and ADAR2. HeLa cells were co-transfected with plasmids encoding GFP-tagged N proteins of MERS-CoV, SARS-CoV-2, bat-CoV, civet-CoV, pangolin-CoV and SARS-CoV or GFP control, and HA-tagged ADAR2. Cell lysates were immunoprecipitated with an anti-GFP antibody, and the expressed proteins were analyzed by western blotting. (B) N proteins from MERS-CoV, SARS-CoV-2, bat-CoV, civet-CoV, pangolin-CoV and SARS-CoV exhibit the ability to co-localize with ADAR1 in SGs in HeLa cells. HeLa cells transfected with various N proteins were treated with AS for 45 min to induce SG formation, followed by immunostaining for N protein, ADAD1 and G3BP1. Scale bar: 20 μm. (C) N proteins from MERS-CoV, SARS-CoV-2, bat-CoV, civet-CoV, pangolin-CoV and SARS-CoV exhibit the ability to co-localize with ADAR2 in SGs in HeLa cells. HeLa cells transfected with various N proteins were treated with AS for 45 min to induce SG formation, followed by immunostaining for N protein, ADAD2 and G3BP1. Scale bar: 20 μm. (D) Quantification of condensates with co-localization of N protein and deaminase in fixed HeLa cells. Data are shown as mean ± SEM ($n = 10$ independent images). Statistical analysis was performed with a one-way ANOVA test. ns > 0.05, ns, not significant.

