## [Peer Review File · The EMBO Journal]

Viral N protein hijacks deaminase-containing RNA granules to enhance SARS-CoV-2 mutagenesis

Zhean Li, Lingling Luo, Xiaohui Ju, Shisheng Huang, Liqun Lei, Jia Liu, Pumin Zhang, Tian Chi, Peixiang Ma, Yanying Yu, Cheng Huang, Xingxu Huang, Qiang Ding, and Yu Zhang

Corresponding authors: Xingxu Huang (huangxx@shanghaitech.edu.cn), Cheng Huang (chuang@shutcm.edu.cn), Qiang Ding (qding@tsinghua.edu.cn), Yu Zhang (zhangy@shanghaitech.edu.cn)

Review Timeline:

Submission Date:	18th Feb 24
Editorial Decision:	12th Mar 24
Revision Received:	28th Apr 24
Editorial Decision:	21st Jun 24
Revision Received:	12th Sep 24
Editorial Decision:	1st Oct 24
Revision Received:	28th Oct 24
Accepted:	6th Nov 24

Editor: Ieva Gailite

Transaction Report:

Dear Xingxu,

Thank you for submitting your manuscript together with the reviews from another journal and your point-by-point response to them to The EMBO Journal. I have now received input from two arbitrating advisors on the revised manuscript. I have copied their comments below.

As you can see, both advisors indicate remaining aspects that would need to be addressed in the revision before I can accept the manuscript for publication here. Specifically, advisor #1 indicates inconsistencies with previous literature regarding the role of ADAR1 vs ADAR2 in antiviral response and their subcellular localisation. This advisor asks to address these discrepancies. From my side, I would appreciate if you could show data on whether viral N protein affects subcellular localisation of ADAR2 and a comparison of ADAR1 vs ADAR2 subcellular localisation in your experimental system. Advisor #2 additionally indicates that further discussion or experimental data on the source of the reported T to C or G to A mutations would be needed.

I would therefore invite you to address these remaining comments in a revised manuscript. I think that it would be useful to discuss the revision in more detail via email or phone/videoconferencing - please let me know which option you prefer.

We generally allow three months as standard revision time. As a matter of policy, competing manuscripts published during this period will not negatively impact on our assessment of the conceptual advance presented by your study. However, please contact me as soon as possible upon publication of any related work to discuss the appropriate course of action. Should you foresee a problem in meeting this deadline, please let us know in advance to discuss an extension.

When preparing your letter of response to the referees' comments, please bear in mind that this will form part of the Review Process File and will therefore be available online to the community. For more details on our Transparent Editorial Process, please visit our website: <https://www.embopress.org/page/journal/14602075/authorguide#transparentprocess>. Please also see the attached instructions for further guidelines on preparation of the revised manuscript.

Please feel free to contact me if have any further questions regarding the revision. Thank you for the opportunity to consider your work for publication, and I look forward to discussing your revision with you.

With best regards,

leva

leva Gailite, PhD
Senior Scientific Editor
The EMBO Journal
Meyerohofstrasse 1
D-69117 Heidelberg
Tel: +4962218891309
i.gailite@embojournal.org

- a point-by-point response to the referees' comments, with a detailed description of the changes made (as a word file).
- a word file of the manuscript text.

- individual production quality figure files (one file per figure)
- a complete author checklist, which you can download from our author guidelines (<https://www.embopress.org/page/journal/14602075/authorguide>).
- Expanded View files (replacing Supplementary Information)
Please see out instructions to authors
<https://www.embopress.org/page/journal/14602075/authorguide#expandedview>

We realize that it is difficult to revise to a specific deadline. In the interest of protecting the conceptual advance provided by the work, we recommend a revision within 3 months (10th Jun 2024). Please discuss the revision progress ahead of this time with the editor if you require more time to complete the revisions.

Referee #1:

This manuscript by Li et al explores the interaction between the SARS-CoV-2 N protein and the APOBEC and ADAR families of RNA editing enzyme. The main point of this manuscript is that the N protein hijacks the deaminases to stress granules to promote RNA editing.

Major points

1. The data presented in this manuscript contradicts published papers so therefore there is an onus on the authors to address these issues. ADAR1 not ADAR2 has been implicated in SARS-CoV-2 infections in human in two publications; Picardi, doi: 10.3390/genes13010041 and Merdler-Rabinowicz doi: 10.1093/nargab/lqad092. In a review by Pfaller et al. doi: 10.1146/annurev-virology-091919-065320 with the exception of one viral infection with BoDV, ADAR1 not ADAR2 is the enzyme is the enzyme involved in the anti-viral response as it is induced by interferon and is an ISG. Therefore, why the authors find that it is ADAR2 not ADAR1 involved, should be addressed. Is it because the N protein is mainly nuclear?
2. ADAR1 is mainly cytoplasmic where as ADAR2 is nuclear. There are many publications on the role of ADAR1 in stress granules, none on the role of ADAR2 in stress granules. In the paper by Weissbach in 2012, doi: 10.1261/rna.027656.111, which was the first to show ADAR1 in stress granules, on page 468 they clearly show that in HeLa cells ADAR2 remains in the nucleus after Arsenite treatment. This is in contradiction to Figure 2b in the manuscript. It is important for the authors to explain why their results are different from the published results. They need to perform the experiments with Arsenite treatment without the N protein to determine if again it is the N protein that is causing the difference.
3. Figure 7 schematic illustration is inaccurate as it does not show the nuclear and cytoplasmic location of the proteins.

Referee #2:

The authors still did not address the core question from reviewer 2: "A to G, T to C, C to T and G to A mutations are detected depending on the ADARs/APOBECs that are present. I do not see how T to C or G to A mutations arise in the plus stranded RNA - there are no known RNA/DNA editing enzymes capable of generating these mutations. These changes must surely be templated by A to G or C to G changes on a complementing (minus) strand".

The authors provided extensive additional data to support important roles of host deaminases in driving viral RNA mutation; however, the sources of T to C and G to A were still not addressed. The reviewer suggested minus strand, which could be a possibility. Additional experiments or at least some literature summary and speculation needs to be added.

Title: N protein hijacks deaminase-involved RNA granules to fuel SARS-CoV-2 mutagenesis

Author: Zhean Li, et al.

Number: EMBOJ-2024-117002-T

A point-by-point response to the reviewer(s)' comments

We thank the reviewers for their insightful comments, which have led to significant improvement of the manuscript. We have performed additional experiments and addressed all points raised by the reviewers. The revised submission has been uploaded, with the main revisions highlighted using track changes mode. Our responses are detailed below.

Reviewer(s)' Comments to Author:

Referee #1:

This manuscript by Li et al explores the interaction between the SARS-CoV-2 N protein and the APOBEC and ADAR families of RNA editing enzyme. The main point of this manuscript is that the N protein hijacks the deaminases to stress granules to promote RNA editing.

Major points

1. The data presented in this manuscript contradicts published papers so therefore there is an onus on the authors to address these issues. ADAR1 not ADAR2 has been implicated in SARS-CoV-2 infections in human in two publications; Picardi, doi: 10.3390/genes13010041 and Merdler-Rabinowicz doi: 10.1093/nargab/lqad092. In a review by Pfaller et al. doi: 10.1146/annurev-virology-091919-065320 with the exception of one viral infection with BoDV, ADAR1 not ADAR2 is the enzyme is the enzyme involved in the anti-viral response as it is induced by interferon and is an ISG. Therefore, why the authors find that it is ADAR2 not ADAR1 involved, should be addressed. Is it because the N protein is mainly nuclear?

Thank you for your insightful comments. The study by Picardi et al. (doi:10.3390/genes13010041) revealed that SARS-CoV-2 infection induces an increase in IFNB1 and ADAR1 expression in Calu-3 cell, suggesting that ADAR1 activity is significantly elevated based on the Alu editing index (AEI). This led to the conclusion that ADAR1 is the primary factor inducing viral mutation (Picardi, Mansi et al., 2022). However, the viral infection only increased IFNB1 and ADAR1 levels in Calu-3 cells and did not activate IFNB1 and ADAR1 in infected Vero cells. The occurrence of A-to-I editing on viral RNA was still observed in infected Vero cells, indicating that activated ADAR1 is not the sole trigger for SARS-CoV-2 genome mutagenesis. Additionally, ADAR2 does contribute to Alu editing and is associated with AEI (Roth, Levanon et al., 2019). Similar to the aforementioned study, the other research (doi: 10.1093/nargab/lqad092) merely demonstrated a correlation between the increased A-to-I editing in host RNA induced by SARS-CoV-2 infection and the enhanced expression of ADAR1. Therefore, these studies do not provide conclusive evidence to confirm that ADAR1, rather than ADAR2, is involved in SARS-CoV-2 mutagenesis.

Some viral infections activate the type-I interferon (IFN-I) response, leading to the increased expression of ADAR1p150, so that ADAR1 is generally considered a major trigger in RNA virus mutagenesis (Borden & Williams, 2011, McNab, Mayer-Barber et al., 2015). However, whether this enhancement is the sole regulator for A to I change in viral RNA is not yet fully understood. Actually, ADAR2 also plays a role in host immune responses and affects RNA virus infection through its deaminase activity, including BoDV, HIV-1, and HDV5 (Doria, Tomaselli et al., 2011, Jayan & Casey, 2002, Tomaselli, Galeano et al., 2015, Yanai, Kojima et al., 2020). Therefore, ADAR1 and ADAR2 share similar characteristics and function as host antiviral factors to regulate viral replication and mutagenesis.

To address your concern, we have conducted a series of experiments, including the impact of ADAR1 on SARS-CoV-2 production and the co-localization of ADAR1 with the N protein in WT and G3BP1/2-null HeLa cells. In fact, both ADAR1 and ADAR2 are involved in the RNA editing of the SARS-CoV-2 genome, although ADAR1 exhibits relatively weaker deaminase activity on the viral genome compared to ADAR2 (Fig. 1H). Moreover, ADAR1 and ADAR2 reduce SARS-CoV-2 production and infection (new Fig. 1I), further confirming their roles in SARS-CoV-2 viral progeny yield and mutagenesis. However, SARS-CoV-2 infection does not alter the expression of ADAR1 and ADAR2 in Caco-2 cells (Fig. EV1C), consistent with the results in Vero cells (Picardi et al., 2022), suggesting that other mechanisms may regulate SARS-CoV-2 mutagenesis. Mechanistically, we found that the N protein can interact with ADAR1 and ADAR2, indicating that these active deaminase enzymes share similar characteristics and are potential host binding partners for the viral N protein (Fig. EV2A). We then transfected N protein mRNA and treated cells with arsenite to promote more N protein and RNA entry into ADAR1/2-localized condensates (Figs. 2B, EV5C, and new EV12B,C). These condensate assemblies facilitate the proximity of viral RNA and N protein to ADAR1/2 (Fig. 2B-D), resulting in an increased number of A>G SNVs in the RNA of N protein (Fig. 2E). Furthermore, we generated G3BP1/2-null cells, transfected N protein mRNA, and treated with stressors. The deletion of G3BP1/2 blocked the formation of ADAR1/2-localized condensates (Figs. 3A and new EV8D), disrupting the spatial proximity between ADAR1/2 and N protein (Figs. 3A and new EV12E), and subsequently failed to enhance the deaminase function on viral RNA (Fig. 3D). Additionally, we introduced extra force to pull viral RNA and N protein into ADAR1/2-involved condensates by adding arsenite to SARS-CoV-2-infected cells. The result was a significant increase in viral genome mutagenesis, both in the number and frequency of A>G/T>C SNVs (Fig. 2G-2I). Thus, ADAR1 and ADAR2 co-regulate A-to-I editing in SARS-CoV-2 mutagenesis.

In addition, as shown by our results and previous studies (Luo, Li et al., 2021, Scherer, Mascheroni et al., 2022, Zheng, Wang et al., 2021), N protein is primarily localized in the cytoplasm and involved in SG formation. ADAR1 and ADAR2 have the ability to participate in SG assembly without N protein expression when cells are exposed to stressors (new Fig. EV8D). Moreover, when N protein is present, it is recruited into ADAR1/2-localized SGs (Figs. 2B, new EV5C, and new

EV12B,C). We have updated the results, figures and discussion in the revised manuscript.

Figure 1. (H) The ADARs mediated SARS-CoV-2 RNA mutagenesis (allelic fraction $\geq 2\%$). Mean with SE were plotted from three biological replicates. The number of SNV with A>G/T>C mutations was normalized to the number of control group. Statistical analysis was performed with unpaired t-test. **(I)** The deaminases regulated SARS-CoV-2 production. Viral genomic RNA expression was normalized to that of the control group. Mean with SE were plotted from six biological replicates. Statistical analysis was performed with one-way ANOVA. ns > 0.05, * P < 0.05, ** P < 0.01, *** P < 0.001, **** P < 0.0001. ns, no significant.

Figure 2. ADAR2 co-localized with N protein in SGs in HeLa cells. The HeLa cells transfected with the N gene and ADAR2 were treated with 200 μ M AS for 45 min to induce SG formation, followed by immunostaining for the N protein, ADAR2 and G3BP1. Scale bar: 20 μ m.

Figure EV5. (C) ADAR1 co-localized with N protein in SGs in HeLa cells. The HeLa cells transfected with the N gene and ADAR1 were treated with 200 μ M AS for 45 min to induces SG formation, followed by immunostaining for the N protein, ADAR1 and endogenous G3BP1. Scale

bar: 20 μ m.

Figure EV8. (D) G3BP1/2 depletion abolished ADAR1/2-localized in SGs when cells are exposed to stressor. The HeLa cells with or without G3BP1/2 dKO were treated with AS for 45 min to induce SG formation, followed by immunostaining for the endogenous G3BP1 and endogenous ADAR1 or ADAR2. Scale bar: 20 μ m.

Figure EV12. (B) N proteins from MERS-CoV, SARS-CoV-2, bat-CoV-2, civet-CoV, pangolin-CoV and SARS-CoV possess the ability to co-localize with ADAR1 in SGs in HeLa cells. The HeLa cells transfected with the various N proteins were treated with AS for 45 min to induce SG formation, followed by immunostaining for N protein, ADAD1 and G3BP1. Scale bar: 20 μ m. **(C)** N proteins from MERS-CoV, SARS-CoV-2, bat-CoV-2, civet-CoV, pangolin-CoV and SARS-CoV possess the ability to co-localize with ADAR2 in SGs in HeLa cells. The HeLa cells transfected with the various N proteins were treated with AS for 45 min to induce SG formation, followed by immunostaining for N protein, ADAD2 and G3BP1. Scale bar: 20 μ m.

Figure EV12. (E) G3BP1/2 depletion abolished the condensate formation of ADAR1 or ADAR2-N proteins from MERS-CoV, SARS-CoV-2, bat-CoV-2, civet-CoV, pangolin-CoV and SARS-CoV. The HeLa cells with G3BP1/2 dKO transfected with the various N proteins were treated with AS for 45 min to induce SG formation, followed by immunostaining for N protein, ADAD1 or ADAR2 and G3BP1. Scale bar: 20 μ m.

2. ADAR1 is mainly cytoplasmic where as ADAR2 is nuclear. There are many publications on the role of ADAR1 in stress granules, none on the role of ADAR2 in stress granules. In the paper by Weissbach in 2012, doi: 10.1261/rna.027656.111, which was the first to show ADAR1 in stress granules, on page 468 they clearly show that in HeLa cells ADAR2 remains in the nucleus after Arsenite treatment. This is in contradiction to Figure 2b in the manuscript. It is important for the authors to explain why their results are different from the published results. They need to perform the experiments with Arsenite treatment without the N protein to determine if again it is the N protein that is causing the difference.

Thank you for your constructive comments. As your mentioned, ADAR1 comprises two isoforms, namely ADAR1p110 and ADAR1p150 respectively. ADAR1p110 is mainly expressed in the nucleus, and ADAR1p150 is capable to move between the nucleus and cytoplasm as a shuttling protein (Shiromoto, Sakurai et al., 2021). Although ADAR2 mostly localizes in both the nucleoplasm and nucleolus, the relatively small amount of ADAR2 is expressed in the cytoplasm (Aizawa, Sawada et al., 2010, Behm, Wahlstedt et al., 2017, Jimeno, Prados-Carvajal et al., 2021, Marcucci, Brindle et al., 2011). To gain further insights into the co-localization of ADAR1 and ADAR2 with N protein in stress-induced RNA granules, we first test the ADAR1 and ADAR2 distribution. When cells are treated with the stressor, a fraction of ADAR1 and ADAR2 moves into

SGs, even without the expression of N protein (new Fig. EV8D). Such involvement of ADAR1 and ADAR2 in SGs is impaired by G3BPs deletion (new Fig. EV8D). The result about ADAR1 is keeping with the reported studies (Sakurai, Shiromoto et al., 2017, Weissbach & Scadden, 2012). Similar to ADAR1, ADAR2 consists with two double-stranded RNA (dsRNA)-binding domains (dsRBDs) and a C-terminal deaminase domain, which is able to bind mRNAs (Matthews, Thomas et al., 2016). SGs are membraneless cell compartments, wherein translation factors, mRNAs, RNA-binding proteins and other proteins assemble together (Yang, Mathieu et al., 2020). Given the low levels of ADAR2 in cytoplasm and the ability to bind mRNA, we therefore considered that ADAR2 is capable to enter into SG assembly. Next, we transfected N protein and sought to determine whether the presence of N protein alter ADAR1 and ADAR2 location. Expression of N protein did not affect the ADAR1 and ADAR2 location, and N protein is recruited into ADAR1/2-existed RNA granules under stress condition due to N protein interaction with G3BP1 and ADAR1/2. Similarly, the absence of G3BP1/2 disrupted the SG assembly, and then abrogated their spatial proximity between ADAR1/2 and N protein (Figs. 2B, new EV5C, and new EV12B-E). Such characteristic and regulation were observed in the N protein from MERS-CoV, bat-CoV, pangolin-CoV, civet-CoV and SARS-CoV (new Fig. EV12B-E). Thus, both ADAR1 and ADAR2 exhibit the ability to associate with N protein and G3BPs in SGs.

In our work, we constructed the vector encoding BFP-tagged ADAR2 to verify whether ADAR2 co-localized with N protein in SGs. The CDS is synthesized based on the longest transcript of ADAR2, which includes the ALU cassette insert and the long C-terminal region. Similar to endogenous expression of ADAR2, it mainly localized in both the nucleoplasm and nucleoli, and only a small fraction is present in the cytoplasm, which is in line with the reported ectopic expression of ADAR2 (Fig. 2B) (Jimeno et al., 2021, Marcucci et al., 2011). The previous study (DOI: 10.1261/rna.027656.111) suggested that ADAR2 is localized exclusively to the nucleoli (Weissbach & Scadden, 2012), which seems to conflict with the aforementioned studies. The discrepancy may be due to the low levels of cytoplasmic ADAR2, making it difficult to detect and potentially overlooking its presence in SGs. Another possibility is that SG disassembly may have occurred. In the methods they employed, transfected cells were treated with sodium arsenite for 30 minutes, followed by a 30-minute recovery period without arsenite. It is important to note that SG formation is a dynamic process, with SGs disassembly and translation resuming upon recovery from stress (Dormann, 2021). Additionally, we constructed a plasmid encoding BFP-tagged ADAR1 and transfected cells with it. The exogenous ADAR1 is predominantly expressed in both the nucleus and cytoplasm and accumulates in SGs under stress conditions (new Fig. EV5C), consistent with the characteristics of endogenous ADAR1. The lack of G3BPs disrupted such involvement of ADAR1/2 in SGs (new Fig. EV8D). The absence of G3BPs disrupted ADAR1/2's involvement in SGs (new Fig. EV8D). Thus, ADAR1 and ADAR2, as RNA-binding proteins, share similar features and possess the ability to participate in SG assembly. We have updated the results, figures and discussion in the revised manuscript.

Figure 2. ADAR2 co-localized with N protein in SGs in HeLa cells. The HeLa cells transfected with the N gene and ADAR2 were treated with 200 μ M AS for 45 min to induce SG formation, followed by immunostaining for the N protein, ADAR2 and G3BP1. Scale bar: 20 μ m.

Figure EV5. (C) ADAR1 co-localized with N protein in SGs in HeLa cells. The HeLa cells transfected with the N gene and ADAR1 were treated with 200 μ M AS for 45 min to induces SG formation, followed by immunostaining for the N protein, ADAR1 and endogenous G3BP1. Scale bar: 20 μ m.

Figure EV8. (D) G3BP1/2 depletion abolished ADAR1/2-localized in SGs when cells are exposed to stressor. The HeLa cells with or without G3BP1/2 dKO were treated with AS for 45 min to induce SG formation, followed by immunostaining for the endogenous G3BP1 and endogenous ADAR1 or ADAR2. Scale bar: 20 μ m.

Figure EV12. (B) N proteins from MERS-CoV, SARS-CoV-2, bat-CoV-2, civet-CoV, pangolin-CoV and SARS-CoV possess the ability to co-localize with ADAR1 in SGs in HeLa cells. The HeLa cells transfected with the various N proteins were treated with AS for 45 min to induce SG formation, followed by immunostaining for N protein, ADAD1 and G3BP1. Scale bar: 20 μm. **(C)** N proteins from MERS-CoV, SARS-CoV-2, bat-CoV-2, civet-CoV, pangolin-CoV and SARS-CoV possess the ability to co-localize with ADAR2 in SGs in HeLa cells. The HeLa cells transfected with the various N proteins were treated with AS for 45 min to induce SG formation, followed by immunostaining for N protein, ADAD2 and G3BP1. Scale bar: 20 μm.

Figure EV12. (E) G3BP1/2 depletion abolished the condensate formation of ADAR1 or ADAR2-

N proteins from MERS-CoV, SARS-CoV-2, bat-CoV-2, civet-CoV, pangolin-CoV and SARS-CoV. The HeLa cells with G3BP1/2 dKO transfected with the various N proteins were treated with AS for 45 min to induce SG formation, followed by immunostaining for N protein, ADAD1 or ADAR2 and G3BP1. Scale bar: 20 μ m.

3. Figure 7 schematic illustration is inaccurate as it does not show the nuclear and cytoplasmic location of the proteins.

Thank you for your kind suggestion. The schematic illustration in Figure 7 has been revised to accurately depict the nuclear and cytoplasmic localization of the proteins, as per your suggestion.

Referee #2:

The authors still did not address the core question from reviewer 2: "A to G, T to C, C to T and G to A mutations are detected depending on the ADARs/APOBECs that are present. I do not see how T to C or G to A mutations arise in the plus stranded RNA - there are no known RNA/DNA editing enzymes capable of generating these mutations. These changes must surely be templated by A to G or C to G changes on a complementing (minus) strand".

The authors provided extensive additional data to support important roles of host deaminases in driving viral RNA mutation; however, the sources of T to C and G to A were still not addressed. The reviewer suggested minus strand, which could be a possibility. Additional experiments or at least some literature summary and speculation needs to be added.

Thank you for pointing out this issue. During viral infection, host deaminases likely perform RNA editing on viral RNA under various scenarios, including 'early' editing on viral genomes and negative-sense transcripts prior to viral replication and 'late' editing on positive/negative-strand transcripts post-replication (Di Giorgio, Martignano et al., 2020). Consequently, A>G/T>C and C>T/G>A transitions are the predominant types of mutations observed in mutational signatures, aligning with previous research (Di Giorgio et al., 2020, Mourier, Sadykov et al., 2021). To exclude the interference of host deaminase-mediated DNA editing on transfected plasmids and to more accurately define the role of deaminases with RNA editing capabilities on viral RNA, we introduced the mRNA of N protein into HeLa cells and performed RNA sequencing. Consistent with the mutational patterns observed in the SARS-CoV-2 genome (Fig. 1A,B), we identified A>G and C>T transitions as the predominant types of mutation in the N protein mRNA (Fig. EV3A,B). G>A and T>C mutations represent the second most common group of changes in the N protein transcript. Such non-classic RNA alterations are also present and documented in mammalian transcripts (Grohmann, Hammer et al., 2010, Klimek-Tomczak, Mikula et al., 2006, Li, Wang et al., 2012, Li, Wang et al., 2011, Niavarani, Currie et al., 2015, Sharma, Bowman et al., 1994, Tao, Ren et al., 2021). Furthermore, APOBEC3A has been reported to regulate G>A mRNA editing in Wilms

Tumor 1 (Niavarani et al., 2015). Transamination and transglycosylation mechanisms have been proposed to be involved in U-to-C editing events in plant transcripts (Castandet & Araya, 2011, Gerke, Szövényi et al., 2020, Knoop, 2023). These studies revealed that there are other regulators to participate in RNA editing process. However, the precise origin of such editing need more investigation due to the lack of a clear molecular mechanism to elucidate these non-classic changes. We have updated the results (line 150-155; line 234-238) and discussion (line 570-577; line 611-631) in the revised manuscript.

References

- Aizawa H, Sawada J, Hideyama T, Yamashita T, Katayama T, Hasebe N, Kimura T, Yahara O, Kwak S (2010) TDP-43 pathology in sporadic ALS occurs in motor neurons lacking the RNA editing enzyme ADAR2. *Acta Neuropathol* 120: 75-84
- Behm M, Wahlstedt H, Widmark A, Eriksson M, Öhman M (2017) Accumulation of nuclear ADAR2 regulates adenosine-to-inosine RNA editing during neuronal development. *Journal of Cell Science* 130: 745-753
- Borden EC, Williams BR (2011) Interferon-Stimulated Genes and Their Protein Products: What and How? *J Interf Cytok Res* 31: 1-4
- Castandet B, Araya A (2011) RNA Editing in Plant Organelles. Why Make It Easy? *Biochemistry-Moscow+* 76: 924-931
- Di Giorgio S, Martignano F, Torcia MG, Mattiuz G, Conticello SG (2020) Evidence for host-dependent RNA editing in the transcriptome of SARS-CoV-2. *Science Advances* 6
- Doria M, Tomaselli S, Neri F, Ciafrè SA, Farace MG, Michienzi A, Gallo A (2011) ADAR2 editing enzyme is a novel human immunodeficiency virus-1 proviral factor. *Journal of General Virology* 92: 1228-1232
- Dormann D (2021) The chains of stress recovery. *Science* 372: 1393-1395
- Gerke P, Szövényi P, Neubauer A, Lenz H, Gutmann B, McDowell R, Small I, Schallenberg-Rüdinger M, Knoop V (2020) Towards a plant model for enigmatic U-to-C RNA editing: the organelle genomes, transcriptomes, editomes and candidate RNA editing factors in the hornwort. *New Phytol* 225: 1974-1992
- Grohmann M, Hammer P, Walther M, Paulmann N, Büttner A, Eisenmenger W, Baghai TC, Schüle C, Rupprecht R, Bader M, Bondy B, Zill P, Priller J, Walther DJ (2010) Alternative Splicing and Extensive RNA Editing of Human Transcripts. *Plos One* 5
- Jayan GC, Casey JL (2002) Increased RNA editing and inhibition of hepatitis delta virus replication by high-level expression of ADAR1 and ADAR2. *Journal of Virology* 76: 3819-3827
- Jimeno S, Prados-Carvajal R, Fernández-Avila MJ, Silva S, Silvestris DA, Endara-Coll M, Domingo-Prim J, Mejías-Navarro F, Rodríguez-Real G, Romero-Franco A, Jimeno-González S, Barroso S, Cesarini V, Aguilera A, Gallo A, Visa N, Huertas P (2021) ADAR-mediated RNA editing of DNA:RNA hybrids is required for DNA double strand break repair. *Nature Communications* 12
- Klimek-Tomczak K, Mikula M, Dzwonek A, Paziewska A, Karczmarski J, Hennig E, Bujnicki JM, Bragoszewski P, Denisenko O, Bomsztyk K, Ostrowski J (2006) Editing of hnRNP K protein mRNA in colorectal adenocarcinoma and surrounding mucosa. *Brit J Cancer* 94: 586-592

Knoop V (2023) C-to-U and U-to-C: RNA editing in plant organelles and beyond. *J Exp Bot* 74: 2273-2294

Li MY, Wang IX, Cheung VG (2012) Response to Comments on "Widespread RNA and DNA Sequence Differences in the Human Transcriptome". *Science* 335

Li MY, Wang IX, Li Y, Bruzel A, Richards AL, Toung JM, Cheung VG (2011) Widespread RNA and DNA Sequence Differences in the Human Transcriptome. *Science* 333: 53-58

Luo LL, Li ZA, Zhao TJ, Ju XH, Ma PX, Jin BX, Zhou YL, He S, Huang JH, Xu X, Zou Y, Li P, Liang AB, Liu J, Chi T, Huang XX, Ding Q, Jin ZG, Huang C, Zhang Y (2021) SARS-CoV-2 nucleocapsid protein phase separates with G3BPs to disassemble stress granules and facilitate viral production. *Sci Bull* 66: 1194-1204

Marcucci R, Brindle J, Paro S, Casadio A, Hempel S, Morrice N, Bisso A, Keegan LP, Del Sal G, O'Connell MA (2011) Pin1 and WWP2 regulate Q/R site RNA editing by ADAR2 with opposing effects. *Embo J* 30: 4211-4222

Matthews MM, Thomas JM, Zheng YX, Tran K, Phelps KJ, Scott AI, Havel J, Fisher AJ, Beal PA (2016) Structures of human ADAR2 bound to dsRNA reveal base-flipping mechanism and basis for site selectivity. *Nature Structural & Molecular Biology* 23: 426-433

McNab F, Mayer-Barber K, Sher A, Wack A, O'Garra A (2015) Type I interferons in infectious disease. *Nature Reviews Immunology* 15: 87-103

Mourier T, Sadykov M, Carr MJ, Gonzalez G, Hall WW, Pain A (2021) Host-directed editing of the SARS-CoV-2 genome. *Biochem Bioph Res Co* 538: 35-39

Niavarani A, Currie E, Reyat Y, Anjos-Afonso F, Horswell S, Griessinger E, Sardina JL, Bonnet D (2015) APOBEC3A Is Implicated in α Novel Class of G-to-A mRNA Editing in Transcripts. *Plos One* 10

Picardi E, Mansi L, Pesole G (2022) Detection of A-to-I RNA Editing in SARS-COV-2. *Genes-Basel* 13

Roth SH, Levanon EY, Eisenberg E (2019) Genome-wide quantification of ADAR adenosine-to-inosine RNA editing activity. *Nat Methods* 16: 1131-+

Sakurai M, Shiromoto Y, Ota H, Song CZ, Kossenkov AV, Wickramasinghe J, Showe LC, Skordalakes E, Tang HY, Speicher DW, Nishikura K (2017) ADAR1 controls apoptosis of stressed cells by inhibiting Staufen1-mediated mRNA decay. *Nature Structural & Molecular Biology* 24: 534-+

Scherer KM, Mascheroni L, Carnell GW, Wunderlich LCS, Makarchuk S, Brockhoff M, Mela I, Fernandez-Villegas A, Barysevich M, Stewart H, Sans MS, George CL, Lamb JR, Kaminski-Schierle GS, Heeney JL, Kaminski CF (2022) SARS-CoV-2 nucleocapsid protein adheres to replication organelles before viral assembly at the Golgi/ERGIC and lysosome-mediated egress. *Science Advances* 8

Sharma PM, Bowman M, Madden SL, Rauscher FJ, Sukumar S (1994) Rna Editing in the Wilms-Tumor Susceptibility Gene, Wt1. *Gene Dev* 8: 720-731

Shiromoto Y, Sakurai M, Minakuchi M, Ariyoshi K, Nishikura K (2021) ADAR1 RNA editing enzyme regulates R-loop formation and genome stability at telomeres in cancer cells. *Nature Communications* 12

Tao J, Ren CY, Wei ZY, Zhang FQ, Xu JY, Chen JH (2021) Transcriptome-Wide Identification of G-to-A RNA Editing in Chronic Social Defeat Stress Mouse Models. *Front Genet* 12

Tomaselli S, Galeano F, Locatelli F, Gallo A (2015) ADARs and the Balance Game between Virus Infection and Innate Immune Cell Response. *Curr Issues Mol Biol* 17: 37-52

Weissbach R, Scadden ADJ (2012) Tudor-SN and ADAR1 are components of cytoplasmic stress granules. *Rna* 18: 462-471

Yanai M, Kojima S, Sakai M, Komorizono R, Tomonaga K, Makino A (2020) ADAR2 Is Involved in Self and Nonself Recognition of Borna Disease Virus Genomic RNA in the Nucleus. *Journal of Virology* 94

Yang P, Mathieu C, Kolaitis RM, Zhang P, Messing J, Yurtsever U, Yang Z, Wu J, Li Y, Pan Q, Yu J, Martin EW, Mittag T, Kim HJ, Taylor JP (2020) G3BP1 Is a Tunable Switch that Triggers Phase Separation to Assemble Stress Granules. *Cell* 181: 325-345 e28

Zheng ZQ, Wang SY, Xu ZS, Fu YZ, Wang YY (2021) SARS-CoV-2 nucleocapsid protein impairs stress granule formation to promote viral replication. *Cell Discov* 7

Dear Xingxu,

Thank you for submitting a revised version of your manuscript. I sincerely apologise for the protracted assessment process due to delays in referee comment submission and protracted discussions within the team.

Your study has now been seen by both original advisors. While advisor #2 now recommends acceptance of the manuscript, advisor #1 remains unconvinced of the reported cytoplasmic localisation of ADAR2. While I appreciate that, as you mention in your point-by-point response, low levels of cytoplasmic ADAR2 do seem to have been reported in several previous publications, I understand the concern by the reviewer that the observation is unusual in the context of the broadly accepted nuclear localisation of ADAR2. To alleviate these concerns, which likely will arise also among expert readers, I would like to ask you to perform the subcellular fractionation experiment requested by advisor #1 while using appropriate nuclear and cytosolic fraction markers. Additionally, please provide an assay testing the specificity of ADAR2 antibody staining of arsenate treatment-induced stress granules in cells with ADAR2 knockdown, which should lead to the loss of ADAR2 stress granule signal. Please note that these experiments are for benefit of the scientific solidity of the study, and I will not return the final revision to the advisors for further evaluation. Finally, please also more clearly highlight in the manuscript that ADAR2 localisation observed in your study is in contrast to previous studies, including that by Weissbach & Scadden (2012), and provide a discussion on potential basis for these differences.

There also are a few editorial points that need addressing in the final version of the manuscript:

1. Please reduce the number of keywords to five.
2. Please remove figures from the manuscript text file.
3. We can accommodate up to five EV figures, which should be uploaded as individual figure files and their legends should be added to the manuscript text after the main figure legends.
4. The remaining EV figures should be added to the "Appendix" together with their legends and renamed "Appendix Figure S1" etc. The figure callouts in the manuscript will need to be updated accordingly.
5. CRediT has replaced the traditional author contributions section because it offers a systematic, machine-readable author contributions format that allows for more effective research assessment. Please remove the Authors Contributions from the manuscript and use the free text boxes beneath each contributing author's name in our online submission system to add specific details on the author's contribution. More information is available in our guide to authors.
6. Please rename "Conflict of interest" section into "Disclosure and competing interests statement" (further info: <https://www.embopress.org/page/journal/14602075/authorguide#conflictsofinterest>).
7. Please update references according to The EMBO Journal style - where there are more than 10 authors on a paper, the first 10 should be listed, followed by 'et al.' Please see further information here: <https://www.embopress.org/page/journal/14602075/authorguide#referencesformat>
8. Please move the Data Availability section to the end of Methods.
9. In the Data Availability section, please add a resolvable link to the PRJNA824251 dataset. More information about the format of this section can be found here: <https://www.embopress.org/page/journal/14602075/authorguide#dataavailability>.
10. In our standard image check we noticed that background signal appears to be missing in the following figure panels: Figure EV8 D, G3BP1/2 dKO samples, G3BP1 staining; Figure EV11 B - N protein/GFP sample, GFP signal, and N protein/ADAR2-RBD1 sample, mCherry signal. Please provide source data for these experiments.
11. In source data for Figure 1I and Figure 6F, we noticed a couple of cases of fully identical values, which appear statistically unlikely (I have attached the corresponding files). A brief explanation how these values were obtained would be very helpful.
12. Our data editors have flagged the following issues in figure legends that need correcting:
 - Please note that in figures 3c-d; 5g; 6e-g; EV 4e; EV 7h-i; there is a mismatch between the annotated p values in the figure legend and the annotated p values in the figure file that should be corrected.
 - Please add information on the nature and number of replicates in the legends of figures EV 1d; EV 7b, f-g.
 - Please describe the nature of replicates in the legends of figures 4c; 5f-g; EV 7a.
 - Please define the error bars in the legends of figures EV 7b, f-g.
13. Papers published in The EMBO Journal are accompanied online by a 'Synopsis' to enhance discoverability of the manuscript. It consists of A) a short (1-2 sentences) summary of the findings and their significance, B) 3-4 bullet points highlighting key results and C) a synopsis image that is 550x300-600 pixels large (width x height, jpeg or png format). You can either show a model or key data in the synopsis image. Please note that the image size is rather small and that text needs to be readable at the final size. Please send us this information together with the revised manuscript.

Thank you again for giving us the chance to consider your manuscript for The EMBO Journal. I look forward to receiving the final revision.

With best wishes,

leva

leva Gailite, PhD
Senior Scientific Editor
The EMBO Journal
Meyerhofstrasse 1
D-69117 Heidelberg
Tel: +4962218891309
i.gailite@embojournal.org

We realize that it is difficult to revise to a specific deadline. In the interest of protecting the conceptual advance provided by the work, we recommend a revision within 3 months (19th Sep 2024). Please discuss the revision progress ahead of this time with the editor if you require more time to complete the revisions.

Referee #1:

Title: N protein hijacks deaminase-involved RNA granules to fuel SARS-CoV-2 mutagenesis Author: Zhean Li, et al. Number: EMBOJ-2024-117002-T
Major concerns

1. I have read this revised manuscript and am still concerned about the issues that I previously address, in my opinion, have not been dealt with. A quick look at all the figures displaying immunofluorescence of ADAR2 in this manuscript, depicts that it is nuclear not cytoplasmic. Obviously if you perform IP in a cell lysate where the nuclear membrane is disrupted then you may observe interactions between ADAR2 and the N-protein. To validate the cytoplasmic interaction of ADAR2 with N protein, a nuclear and cytoplasmic fractionation should be performed before the IP to show the localization of ADAR2, either nuclear or cytoplasmic.

The authors claim that ADAR2 is mainly nuclear, so how much is cytoplasmic? Is it inconsequential?
Can they measure it?

2 The authors argue that ADAR2 is present in low abundance in the cytoplasm and quote these manuscripts (Aizawa, Sawada et al., 2010, Behm, Wahlstedt et al., 2017, Jimeno, Prados-Carvajal et al., 2021, Marcucci, Brindle et al., 2011). I checked and none of these manuscripts support this claim. I recommend that the authors read these manuscripts as they claim the opposite, in particular Behm 2017 where they demonstrate that there is an NLS between residues 48-72 of ADAR2. This explains the deletion series of ADAR2 they obtain in Figure 5, as many of the deletions mis-localize ADAR2 and this is obvious from the immunofluorescence in this figure.

3 As I previously mentioned the manuscript by Weissbach and Scadden is a seminal publication in this field. The authors never discuss in this manuscript why their results differ from those published. They say in their response to the reviewers that it could be the length of time of arsenate treatment, then this has to be proven. Quite frankly the HeLa cells in Figure 6B of the Weissbach paper treated with arsenate look much better than Figure 2B of this manuscript. There are 14 manuscripts describing ADAR1 in stress bodies, none describing ADAR2. Yet this point is never address in this revised manuscript.

4 The schematic illustration in Figure 7 regarding ADARs is completely inaccurate as it depicts more ADARs in the cytoplasm than in the nucleus. The ADARs should be depicted as either ADAR1 or ADAR2 as they have different localizations.

Referee #2:

The authors provided reasonable response to my comments. I am ok with publishing the work.

Title: N protein hijacks deaminase-involved RNA granules to fuel SARS-CoV-2 mutagenesis

Author: Zhean Li, et al.

Number: EMBOJ-2024-117002R

A point-by-point response to the editor(s)' comments

We thank the editor and reviewers for insightful comments, which have led to significant improvement of the manuscript. We have performed additional experiments and addressed all raised concerns. The revised submission has been uploaded, with the main revisions highlighted using track changes mode. Our responses are detailed below.

Editor(s)' Comments to Author:

Your study has now been seen by both original advisors. While advisor #2 now recommends acceptance of the manuscript, advisor #1 remains unconvinced of the reported cytoplasmic localisation of ADAR2. While I appreciate that, as you mention in your point-by-point response, low levels of cytoplasmic ADAR2 do seem to have been reported in several previous publications, I understand the concern by the reviewer that the observation is unusual in the context of the broadly accepted nuclear localisation of ADAR2. To alleviate these concerns, which likely will arise also among expert readers, I would like to ask you to perform the subcellular fractionation experiment requested by advisor #1 while using appropriate nuclear and cytosolic fraction markers. Additionally, please provide an assay testing the specificity of ADAR2 antibody staining of arsenate treatment-induced stress granules in cells with ADAR2 knockdown, which should lead to the loss of ADAR2 stress granule signal. Please note that these experiments are for benefit of the scientific solidity of the study, and I will not return the final revision to the advisors for further evaluation. Finally, please also more clearly highlight in the manuscript that ADAR2 localisation observed in your study is in contrast to previous studies, including that by Weissbach & Scadden (2012), and provide a discussion on potential basis for these differences.

Thank you for your insightful comments. We also thank the favorable comments from advisor#2 and the constructive comments from advisor#1. We greatly appreciate that the concerns regarding the distribution of ADAR2 are very critical to our study. Per your suggestion, we conducted subcellular fractionation assay, followed by immunoblotting detection. We used GAPDH as a marker for cytoplasmic proteins and Lamin A/C as a marker for nuclear proteins, respectively. As expected, both exogenous and endogenous ADAR2 were predominantly found in the nucleoplasm and nucleolus, with a relatively minor fraction present in the cytoplasm (Appendix Fig. 6A,B). These findings regarding ADAR2 distribution are consistent with our immunofluorescence results and are in accordance with the previous studies (Aizawa *et al*, 2010; Behm *et al*, 2017; Jimeno *et al*, 2021; Marcucci *et al*, 2011). Additionally, we knocked down ADAR2 expression in HeLa cells and conducted immunofluorescence staining with anti-ADAR2 antibody to evaluate its specificity.

Quantitative analysis of the total immunofluorescence intensity revealed a significant reduction of 85% in fluorescence intensity following shRNA-mediated knockdown of ADAR2 (Appendix Fig. 6C,D). Furthermore, the fluorescence intensity within stress granules was greatly reduced (Appendix Fig. 6C,F,G). These decreases confirm the antibody's specificity for ADAR2 and further support the presence of ADAR2 in cytoplasm. We have updated the results (line 320-335) and discussion (line 677-698) in the revised manuscript.

Appendix Figure S6. ADAR2 mainly localized in both the nucleoplasm and nucleoli, and only a small fraction is present in the cytoplasm. (A) The endogenous ADAR2 distribution in cytoplasm and nucleus. Cell lysates were fractionated to separate the cytosol and the nuclei, and were immunoprecipitated with an anti-ADAR2 antibody. GAPDH and lamin A/C proteins were used as loading control in the cytoplasmic and nuclear fraction, respectively. **(B)** The exogenous ADAR2 distribution in cytoplasm and nucleus. HeLa cells were transfected with plasmids encoding mCherry-tagged ADAR2. Cell lysates were fractionated to separate the cytosol and the nuclei, and were immunoprecipitated with an anti-mCherry antibody. GAPDH and lamin A/C proteins were

used as loading control in the cytoplasmic and nuclear fraction, respectively. **(C)** ADAR2 co-localized with N protein in SGs in response to AS. HeLa cells were transfected with ADAR2-shRNA or control and then treated with AS for 45 min to induce SG formation, followed by immunostaining for ADAR2 and G3BP1. Scale bar: 10 μm . **(D)-(E)** Quantitative analysis of the total fluorescence intensity (D) and mean fluorescence intensity (E) for ADAR2 related to panel (C). Data are shown as means \pm SEM (n = 20 independent images). Statistical analysis was performed with one-way ANOVA. ****P < 0.0001. **(F)-(G)** Quantitative analysis of the total fluorescence intensity (F) and mean fluorescence intensity (G) for per stress granule related to panel (C). Data are shown as means \pm SEM (n = 130~140 independent stress granule). Statistical analysis was performed with two-tailed unpaired t-test. ****P < 0.0001.

References

Aizawa H, Sawada J, Hideyama T, Yamashita T, Katayama T, Hasebe N, Kimura T, Yahara O, Kwak S (2010) TDP-43 pathology in sporadic ALS occurs in motor neurons lacking the RNA editing enzyme ADAR2. *Acta Neuropathol* 120: 75-84

Behm M, Wahlstedt H, Widmark A, Eriksson M, Öhman M (2017) Accumulation of nuclear ADAR2 regulates adenosine-to-inosine RNA editing during neuronal development. *Journal of Cell Science* 130: 745-753

Jimeno S, Prados-Carvajal R, Fernández-Avila MJ, Silva S, Silvestris DA, Endara-Coll M, Domingo-Prim J, Mejías-Navarro F, Rodríguez-Real G, Romero-Franco A *et al* (2021) ADAR-mediated RNA editing of DNA:RNA hybrids is required for DNA double strand break repair. *Nature Communications* 12

Marcucci R, Brindle J, Paro S, Casadio A, Hempel S, Morrice N, Bisso A, Keegan LP, Del Sal G, O'Connell MA (2011) Pin1 and WWP2 regulate Q/R site RNA editing by ADAR2 with opposing effects. *Embo J* 30: 4211-4222

Dear Xingxu,

Thank you for submitting a revised version of your manuscript and for addressing the remaining editorial and experimental points. While everything appears in the order now, I noted that the manuscript requires in-depth editing for scientific English language. Please involve either an appropriate scientific language editing service or a colleague that is a native English speaker.

I have also gone through the title, abstract and synopsis of the manuscript, and would like to propose the edits included below and in the attached file. I have also written a short blurb that will accompany the title of your manuscript on our online page of contents. Please take a look and let me know if any corrections are needed.

Title:

Viral N protein hijacks deaminase-containing RNA granules to enhance SARS-CoV-2 mutagenesis

Blurb:

Coronaviral N protein interacts and co-condensates with host ADAR and APOBEC deaminases to promote viral RNA editing.

Synopsis:

Host cell-encoded deaminases act as innate restriction factors that enhance viral genome mutation and evolution. This study shows that SARS-CoV-2 nucleocapsid (N) protein interacts with host deaminases at RNA granules, thus promoting viral mutagenesis.

- N protein specifically interacts with host ADAR and APOBEC deaminases.
- N protein localises to deaminase-containing RNA granules via RNA-dependent phase separation.
- N protein promotes deaminase-induced RNA editing of SARS-CoV-2 genome.
- N protein with a F17A mutation fails to localise to deaminase-containing RNA granules and reduces host deaminase-dependent mutagenesis of SARS-CoV-2 genome.

In the synopsis image, I would recommend changing the description of the step 2 to: "Deaminase-containing condensate formation".

Please feel free to contact me if have any questions regarding this final revision. Thank you again for giving us the chance to consider your manuscript for The EMBO Journal. I look forward to receiving the revised version.

With best regards,

Ieva

We realize that it is difficult to revise to a specific deadline. In the interest of protecting the conceptual advance provided by the work, we recommend a revision within 3 months (30th Dec 2024). Please discuss the revision progress ahead of this time with the editor if you require more time to complete the revisions.

The authors addressed the remaining editorial issues.

Dear Xingxu,

Thank you for addressing the final editorial points. I am now pleased to inform you that your manuscript has been accepted for publication at The EMBO Journal.

Finally, we would like to promote your manuscript among the Chinese readership. Therefore, we would like to invite you to prepare a short summary of the manuscript in Chinese (1500-2000 Chinese characters), which we will promote on the WeChat platform 'BioArt' with more than 610,000 followers.

If you are interested in this opportunity, we recommend covering the article very close to its online publication date. Thus, ideally we would very much appreciate if you could send us a draft within the next 7 working days. Please let us know whether or not you would be interested in contributing such a short summary in Chinese.

I have included below some general guidelines on how to prepare a summary and a link to recent examples for your reference. Please let me know if you have any questions about this.

If you have any questions, please do not hesitate to contact the Editorial Office. Thank you for this contribution to The EMBO Journal and congratulations on a nice study!

With best wishes,

Ieva

General WeChat Summary Guidelines

1. These summary articles are meant to be targeting general audience so please limit the use of specialized technical terms, acronyms and jargon.
2. A summary usually starts with brief background information of the reported work, which is followed by explaining the findings in some detail, and ends with a short review of the conclusions as well as the implications of the work and future directions for the research.
3. The summary should at least contain one graphical item, such as a scheme or a figure from the paper.
4. Please provide ONE SINGLE document containing all text and graphical materials, ideally as a Word.docx or .doc file. Please DO NOT provide the document as a .pdf file.
5. Please DO NOT publicly release the document before the paper is officially published online.

Summary Examples

EMBO J | 罗招庆/欧阳松应揭示谷酰胺脱氨酶MvcA的去泛素化功能
